A complete, multi-level conformational clustering of antibody complementarity-determining regions

Nikoloudis Dimitris 1
Pitts Jim E. 1
Saldanha José W. 2 jsaldan@nimr.mrc.ac.uk
1 Department of Biological Sciences, Birkbeck College, University of London , London , UK
2 Division of Mathematical Biology, National Institute for Medical Research , London , UK
Martens Lennart
Electronic publication date: 2014 Jul 1
Publication date: 2014
Volume: 2
Electronic Location ID: e456
Received 2014 Mar 23; Accepted 2014 Jun 5
Copyright: © 2014 Nikoloudis et al.
Copyright year: 2014
Copyright holder: Nikoloudis et al.
License: This is an open access article distributed under the terms of the Creative Commons Attribution License, which permits unrestricted use, distribution, reproduction and adaptation in any medium and for any purpose provided that it is properly attributed. For attribution, the original author(s), title, publication source (PeerJ) and either DOI or URL of the article must be cited.
License URL: https://creativecommons.org/licenses/by/4.0/

Keywords: Antibody structure, Canonical model, CDR conformation, Dynamic hybrid tree-cut, Humanisation, Clustering, Nested architecture, Redundant repertoire, Prediction

Funding: The authors declare no external funding sources.

==============================
Classification of antibody complementarity-determining region (CDR) conformations is an important step that drives antibody modelling and engineering, prediction from sequence, directed mutagenesis and induced-fit studies, and allows inferences on sequence-to-structure relations. Most of the previous work performed conformational clustering on a reduced set of structures or after application of various structure pre-filtering criteria. In this study, it was judged that a clustering of every available CDR conformation would produce a complete and redundant repertoire, increase the number of sequence examples and allow better decisions on structure validity in the future. In order to cope with the potential increase in data noise, a first-level statistical clustering was performed using structure superposition Root-Mean-Square Deviation (RMSD) as a distance-criterion, coupled with second- and third-level clustering that employed Ramachandran regions for a deeper qualitative classification. The classification of a total of 12,712 CDR conformations is thus presented, along with rich annotation and cluster descriptions, and the results are compared to previous major studies. The present repertoire has procured an improved image of our current CDR Knowledge-Base, with a novel nesting of conformational sensitivity and specificity that can serve as a systematic framework for improved prediction from sequence as well as a number of future studies that would aid in knowledge-based antibody engineering such as humanisation.

Introduction

Antibodies achieve the recognition and binding of antigens mainly by variation in the length and sequence of six loops called complementarity-determining regions (CDRs), three in the Light chain (CDR-L1, -L2, -L3) and three in the Heavy chain (CDR-H1, -H2, -H3). Early comparison of the experimental data suggested that CDRs usually adopt one of a limited number of possible conformations, depending on the presence of a few key residues in the sequence. This observation gave rise to the canonical model in which the three-dimensional conformation (or canonical class) of the corresponding loop could be predicted from sequence templates for five of the six CDRs (Chothia et al., 1986; Chothia et al., 1989; Chothia et al., 1992; Chothia & Lesk, 1987). Since this initial classification, further analysis has revealed novel classes, improved the predictability of the known ones, and offered insights into antigen recognition and binding mechanisms (Martin & Thornton, 1996; Al-Lazikani, Lesk & Chothia, 1997). Later, a number of studies (Shirai, Kidera & Nakamura, 1996; Shirai, Kidera & Nakamura, 1999; Furukawa et al., 2001; Kuroda et al., 2008) provided structure-determining sequence rules for the prediction of the base conformation of the sixth and final CDR-H3.

Today, the increasing amount of new structural data presents an opportunity not only to improve the accuracy of conformational prediction from sequence alone, by identifying novel classes and reassessing the known ones; but also to study the basis of loop folding and gain insights into subtle antibody/antigen interactions. Steps are being taken in this direction that will enhance the capabilities of knowledge-based antibody engineering, e.g., humanization (Saldanha, 2009) and assist attempts at de novo antibody design (Yu et al., 2012). In this study, an updated repertoire of CDR conformations was acquired by clustering and analysis of all available antibody loop structures. The primary goal was to create a complete repository of the redundant CDR conformational repertoire that is observed and deposited in the Protein Data Bank (PDB, Berman et al., 2000), i.e., obtain a classification for every single CDR, regardless of quality or sequence redundancies. This would allow a number of better informed, dedicated analyses regarding sequence-to-structure relations, induced fit, structural consistency, mutation studies or more targeted thermodynamic simulations. Most previous work was conducted when only a limited number of structures were available (Chothia et al., 1989; Martin & Thornton, 1996; Barré et al., 1994; Rees et al., 1994; Reczko et al., 1995; Tomlinson et al., 1995; Morea et al., 1997; Guarne et al., 1996; Morea et al., 1998; Morea, Lesk & Tramontano, 2000; Oliva et al., 1998), or only specific CDRs were targeted for clustering (Kuroda et al., 2009; Teplyakov & Gilliland, 2014), or the selected datasets were heavily filtered in order to avoid redundancies and the inclusion of potentially wrong structures (North, Lehmann & Dunbrack, 2011). The automatically updated online repertoire AbYsis is maintained at http://www.bioinf.org.uk/abysis, however it doesn’t annotate the redundant CDR content. In contrast, the very recently released CDR structural database SAbDab (Dunbar et al., 2014) does contain the redundant CDR repertoire, but the characteristics of the clustering method employed are very different from the present work, as indicated later.

A strategic decision was made to include all redundant CDR conformations, especially those from the same antibody presented in different PDB structure files and those from multiple copies of the same antibody variable chain within the same PDB file. Previous experience with examining CDR conformations suggested that different structures or copies of the same CDR may reveal its conformational flexibility, which is a useful aspect for molecular modellers and biologists who study the antigenic interface. By randomly selecting only one structure file and one variable chain copy of a given CDR, there is the risk of picking a non-representative instance which is different from the CDR’s average conformation, or picking a structure that contains errors or invasive crystal packing. Furthermore, random selection also removes from the dataset the possibility of observing an antibody in both its free and bound state, wherever this is available. Finally, it was judged that a poor average crystallographic resolution does not a priori point to a wrong structure and that a corresponding pre-filtering would potentially prevent the inclusion of new conformations in the repertoire.

The second goal was to take advantage of all antibody structural information in order to create CDR clusters that can lead to advancement in the area of conformational prediction from sequence alone (Nikoloudis, Pitts & Saldanha, 2014). The enrichment of the cluster populations (CDRs with the same or similar conformations) with as many examples as possible is crucial to allow the making of connections between sequence and structure. The present analysis aimed to serve as a preliminary framework not only by producing an updated conformational dataset, but also by creating a novel nested clustering architecture that is more beneficial for prediction from sequence alone. Specifically, the nested repertoire tries to optimise the trade-off between the proliferation of sequence examples and a possible detrimental effect from small structure-solving errors.

By including all available CDR structures in the dataset, any conclusions on conformational validity were shifted to the post-clustering stage of analysis. However, at the same time there is an increase in noise of the dataset and as a consequence it was expected that the extents of some of the natural conformational clusters could be distorted or overlapping. These characteristics were taken into consideration in the design of the clustering steps in order to optimise the cluster separation, while minimising the loss of cluster specificity and/or sensitivity. The clustering procedure itself should help with the assessment of conformational validity and act as a first filter by efficiently excluding outliers from the natural clusters.

Methods

Acquisition of antibody structure files

The three-dimensional coordinates of all antibody structures were downloaded from the PDB (Berman et al., 2000). Since the presence of antibody variable chains inside a PDB file is not annotated in a unique and systematic way, the advanced search tool of the database was used in order to apply composite search filters. The simple text search query of the database with the keywords “antibody” or “immunoglobulin” returns hundreds of unwanted PDB files, for example those that only contain a constant antibody fragment (Fc) or those that contain the keyword in their primary citation without any relevant structures in the file. Conversely, in several cases, antibody variable chains (Fv) are found in PDB files that do not contain the keywords “antibody” or “immunoglobulin” at all. In order to refine the obtained results, multiple queries were run using a variety of relevant keywords and their combinations with appropriate logical AND/OR/NOT connectors. The keywords employed typically included: “antibody”, “immunoglobulin”, “Fab”, “Fv”, “Fc”, “light chain”, “heavy chain”, “intact”, “complete”, “camelid”, “llama”, “VHH”, “light dimer” and “Bence -Jones”.

The final dataset comprised of exactly 1,351 PDB structure files, 8 of which contain variable chains from two different antibodies (5 were idiotype-anti-idiotype complexes), increasing the total number of antibody structures to 1,359. The total number of included CDRs is 12,712, 2,827 of which are unique in sequence. Table 1 contains a summary of the dataset contents. The dataset was locked on the 31st of December 2011 and should reflect the complete repertoire of antibody CDR structures up to that date. The set should be complete, given the proviso that there was a lack of specific tagging or annotation in the required PDB files.

Table 1 Summary of clustering dataset contents.

Total clustered members per CDR include outliers and singletons.

Total PDB files	1,351	
Files containing structures from two
antibodies/idiotypes-anti-idiotypes	8/5	
Total antibody structures	1,359	
Total number of CDRs	13,086	
CDRs with missing Cα coordinates	374	
Total clustered CDRs	12,712	
CDR-L1 clustered	2,155	
CDR-L2 clustered	2,174	
CDR-L3 clustered	2,164	
CDR-H1 clustered	2,057	
CDR-H2 clustered	2,130	
CDR-H3 clustered	2,032	
Total non-redundant CDR sequences	2,827	
PDB files with lambda isotypes	194	
Heavy only	77	
Light only	78	
PDB files with bound antibodies	673	

Numbering of antibody variable chains and definition of CDR extents

All the antibody variable chain sequences in the dataset were structurally numbered in order to detect the beginning and end of each CDR, using regular expressions for the detection of the location of conserved sequence patterns. The initially adopted numbering scheme was the Chothia scheme (Chothia & Lesk, 1987) because it correctly places the insertion points in CDR-L1 and CDR-H1, but also because it is very frequently used in the CDR-related literature. The definitions used for the extents of CDRs-L1, -L2, -L3 and -H3 were also those established by Chothia & Lesk (1987) because they are most commonly used. However, for CDR-H1 and CDR-H2, the definitions adopted were those used in North, Lehmann & Dunbrack (2011). Based on previous experience from the visual examination of CDR-H1 structural superpositions, it was noted that the N-terminal portion of the loop where Kabat’s (Kabat et al., 1991) and Chothia’s CDR-H1 differ shows great variability both in sequence and structure. Thus, it was judged that this cluster analysis would be more revealing and useful if the CDR-H1 extent was considered as the entire length of the loop, namely residues H23–H35. As far as CDR-H2 was concerned, it was observed that the C-terminal portion of Kabat’s definition (i.e., residues H59–H65) remained relatively unchanged conformationally in most CDRs. Therefore, only the length of the symmetrical loop portion between residues H50–H58 was retained for the CDR-H2 definition.

CDR length and numbering scheme amendments

A number of antibodies contained a CDR with more residues than the current scheme could accept. The CDRs concerned were CDR-L2, -L3, -H1, -H2 and -H3. These CDRs, except for CDR-L2, already contained an insertion locus so the maximum allowed length was extended by adding more insertion positions (letters) to the numbering scheme. An insertion point was required in CDR-L2 for an 11-residue length. By superposing the new 11-residue loop (PDB code 3FFD) on a typical 7-residue one (1A4K), it was strongly suggested that the insertion point in CDR-L2 should be placed at position L51 (Fig. 1).

Figure 1 Superposition of 7-residue and 11-residue CDR-L2.

The 5 C-terminal residues of 1A4K (in red) 7 residue CDR-L2 (L52–L56) are superposed to the equivalent portion of 3FFD (in blue) 11 residue CDR-L2. Position L51 is highlighted in green, as the best insertion point in the structural numbering scheme. Graphics created with Swiss-PdbViewer (http://www.expasy.org/spdbv/).

Two more cases required intervention in the numbering scheme. The first was in Light chain framework-3 (LFR3), where structure 1PW3 showed a 2-residue insertion. Superposition of this structure to the respective portion of a typical Light variable chain (1A4K) revealed that an insertion point should be introduced at position L67 (Fig. 2). The second case was raised by two anti-HIV antibodies observed in structures 3RPI and 3SE8, showing an insertion of 3 and 7 residues respectively in Heavy chain framework-3 (HFR3). Superposition of these frameworks onto a typical HFR3 (3MLY) suggested that an insertion point should be placed at residue H74 (Fig. 3). Table 2 summarises all the amendments brought to the initial numbering scheme in order to accommodate the special cases discovered in the dataset.

Figure 2 Superposition of Light Framework 3 with an insertion onto a typical LFR3.

Residues L60–L75 of crystal structure 1PW3 (in red), containing an insertion, are superposed onto a typical example of the equivalent Light chain fragment (here 1A4K, in blue). The new insertion point was introduced in position L67 (highlighted in green). Graphics created with Swiss-PdbViewer (http://www.expasy.org/spdbv/).

Figure 3 Superposition of Heavy Framework 3 with an insertion onto a typical HFR3.

The Cα-trace of a two-leg superposition of residues H65–H73 and H76–H78 of crystal structures 3RPI (in yellow) and 3SE8 (in red), containing an insertion, onto the equivalent residues of a typical structure without an insertion (here 3MLY, in blue). The proposed insertion point H74 is highlighted in green in 3MLY and is shown with its side chain (Ser). Graphics created with Swiss-PdbViewer (http://www.expasy.org/spdbv/).

Table 2 Modifications brought to the numbering scheme.

Modifications brought to the numbering scheme in the light of new and atypical sequences. LFR3, light chain framework 3; HFR3, heavy chain framework 3. CDR-H3 insertion positions H100uvw were not required in the present dataset, but were added for the technical continuity up to the pre-existing positions H100xyz and for future use. Thus 3U1S has a CDR-H3 length of 31 residues.

Locus	Numbering
scheme addition	Maximum
CDR length	Structures with the
new maximum length	CDR
extents	
CDR-L1	–	17	N/A	L24-L34	
CDR-L2	L51abcd	11	2GSG, 2H32, 2H3N, 2OTU,
2OTW, 2QHR, 3FFD	L50-L56	
LFR3	L67ab	N/A	1PW3	N/A	
CDR-L3	L95cd	13	2GSG, 2OTU, 2QHR, 3FFD,
3MLW	L89-L97	
CDR-H1	H31cdefghijk	24	3K3Q	H23–H35	
CDR-H2	H52ef	15	3TWC, 3TYG	H50–H58	
HFR3	H74abcdefg	N/A	3SE8	N/A	
CDR-H3	H100nopqrstuvw	34	3U1S	H95-H102	

Clustering overview

In order to increase the usefulness of the clustering result in a way that meets the needs of a wider range of applications, a novel three-level nested cluster architecture was devised. At the parent-level, members of the same cluster share the least similarity in terms of Cα-atom Root-Mean-Square Deviation (RMSD), as the cluster is designed to include all the variants of a conformational theme within the limits of a statistical cluster validation. At the daughter-level, RMSD variance is successively reduced and members of the same cluster are increasingly similar. This stratified scheme could also be perceived as a variation of sensitivity to the potential natural flexibility of a CDR conformation (looser clusters), as well as a trade-off to the specificity of a particular shape (tighter clusters).

First-level clusters were formed by the use of a statistical clustering method, while second- and third-level clusters were defined using qualitative criteria. More specifically, the data was initially analysed by average- and complete-distance hierarchical clustering using RMSD distance matrices, and pruning of the resulting trees was performed with the Dynamic Tree Cut algorithm (Langfelder, Zhang & Horvath, 2007). RMSD distance matrices were obtained by performing all-by-all Cα-atom superpositions of the entire CDR loops, per individual CDR length. The result of hierarchical clustering was a set of level-1 structural classes, as traditionally produced by various methods in all previous CDR conformational studies, meaning that members of the same cluster were similar to a degree that is defined by the tree-pruning and clustering criteria.

Subsequently, ϕ/ψ angles were calculated for all CDR residues, each residue was attributed to a Ramachandran region and Ramachandran logos were formulated for each CDR. For practical and computational reasons, the boundaries of the different Ramachandran regions were based on the Ramachandran Plot subdivision used by North, Lehmann & Dunbrack (2011) (Fig. 4). Two types of Ramachandran logos are defined for each CDR, namely one where similar conformational regions were represented by the same letter (also suggested in North, Lehmann & Dunbrack, 2011), which will henceforth be called the reduced-Ramachandran Logo or r-RL, and one where every conformational region is represented individually, called the full-Ramachandran Logo or f-RL. For the formation of level-2 clusters, the members of any given parent level-1 cluster were regrouped by identical r-RL, meaning that members of the same cluster contain residues at each CDR position that belong to similar conformational regions. For the formation of level-3 clusters, the members of any given level-2 cluster were regrouped by identical f-RL, meaning that members of the same cluster contain residues at each CDR position that belong to the exact same conformational region. An example showing the layout of this nested cluster architecture can be seen in Fig. 5. Outliers/singletons were all given the tag ‘-O-‘ in their conformational logo, which created a common parent class that allowed the subsequent formation of 2nd- and 3rd-level clusters within outlier space, as well.

Figure 4 Ramachandran plot divided into conformational regions.

A: α-helix region; B: β-sheet region; D: δ-region; G: γ-region; L: left-handed helix region; P: polyproline II region. For the construction of reduced-Ramachandran logos (r-RL), residues belonging to regions with similar conformations were represented by the same letter: (A/D) = A, (B/P) = B, (L/G) = L. For the construction of full-Ramachandran logos (f-RL), each conformational region was represented individually. E.g., Ramachandran logos for CDR-L3 1TJH_L:r-RL: BBAABBBBB f-RL: BBDABPPPB.

Figure 5 Example of the nested clusters architecture.

Level-1 cluster H1-13-III (i.e., the third top-level cluster of 13-residues CDR-H1), defined by RMSD-based hierarchical clustering, contains 3 Level-2 clusters, the members of each sharing the same reduced-Ramachandran logo, and in total 11 Level-3 clusters, the members of each sharing the same full-Ramachandran logo. All Level-3 clusters share the same reduced-Ramachandran logo with their parent Level-2 cluster, but each one displays a distinct full-Ramachandran logo.

Clustering method

The RMSD distance matrices produced for each CDR/length combination were used for hierarchical analysis in the statistical package RGui (GNU project, http://www.sciviews.org/_rgui/). The average-linkage and complete-linkage algorithms were preferred to single-linkage in order to avoid chaining effects in dense configurations of the dataset in conformational space, and were both explored for every CDR/length combination. Hierarchical trees (dendrograms) that gave a Cophenetic Correlation Coefficient (CPCC) lower than 0.6 were directly discarded as pointing to poor fitting of the data. In all cases at least one of the hierarchical methods achieved a CPCC score greater than 0.6. Both hierarchical trees were considered whenever the CPCC was acceptable and comparatively evaluated using the criteria below.

The Dynamic Hybrid Tree Cut method of the Dynamic Tree Cut statistical package in RGui was utilised for dendrogram pruning. The package has been previously successfully used for the detection of biologically meaningful clusters in a protein–protein interaction network in Drosophila (Dong & Horvath, 2007). The Dynamic Hybrid Tree Cut algorithm offers flexibility, by allowing the user to set the desired pruning parameters for cluster and outlier recognition. Specifically, the algorithm defines four cluster shape criteria: (1) the minimum number of cluster members (N0, minClusterSize), (2) the maximum scatter of the pairwise distances between the lowest merged objects (CDR structures) in each cluster, called the cluster core (dmax, maxAbsCoreScatter), (3) the maximum joining height at which a cluster attaches to the rest of the dendrogram (hmax, cutHeight), and (4) the minimum distance between the core scatter and the joining height of a cluster to the dendrogram, called the cluster gap (gmin, minAbsGap). The core scatter is defined as the average of all pairwise dissimilarities between objects belonging to the core of the cluster. Consequently, a branch is considered a cluster when it contains a minimum number of members (N0), its joining height is at most hmax, its core is tightly connected (dmax) and distinct from its neighbourhood (gmin). Specifically, the minimum cluster gap distance (gmin) can be perceived as the minimum allowance for the cluster to expand its diameter from its core until it reaches a neighbouring cluster.

Although these pruning parameters are explained in depth in the corresponding method paper (Dong & Horvath, 2007), an example of the application of pruning parameters to an actual dendrogram from this analysis can be seen in Fig. 6. The number of objects assigned to the core of a cluster is derived from the following implemented formula: (1) nc=min∫N0/2+N−N0/2,N

with nc the number of core objects, N0 the defined minimum cluster size and N the total number of objects in the cluster. As a consequence, the core of small clusters can be as large as the whole cluster, while the core of large clusters remains a fraction of the lowest joined objects.

Figure 6 Illustration of the parameters taken into account for the dendrogram pruning of CDR-L1/12 residues with the Dynamic Hybrid method. The minimum gap statistic (gmin) defines the minimum required distance between the average core scatter and the joining height of the clusters (‘Gap’), for successful cluster formation. In this example, gmin is set lower than the displayed Gaps, so nodes above its value were considered as different clusters.

The algorithm examines the dendrogram in a bottom-up manner and attempts to perform three types of branch merges: a merge of two singletons which creates a new branch, the addition of a singleton to a branch, or a merge of two branches. In each step two branches are tested against the pruning criteria: if both considered branches satisfy the criteria then both are declared “closed” and no further objects are added in the current step. Otherwise, the branches are merged and this new group is reassessed for cluster conformity during the next merge with an adjacent branch. Objects too far from a cluster are left unlabelled as outliers. Once all possible object assignments are performed, the method allows a further optional ‘Partitioning Around Medoids-like’ step (PAM). During this step, unlabelled objects (outliers) are considered one-by-one and are assigned to existing clusters based on a user-defined maximum allowable distance, or when their distance is smaller than the cluster’s radius. There are two options available for the cluster radius definition (parameter: useMedoids[=FALSE/TRUE]). If average distances are being used (FALSE), then the radius of the cluster is defined as the maximum of the average distances between objects in the cluster. If instead medoids are used (TRUE), then the radius is defined as the maximum distance of the cluster’s medoid to the cluster’s objects.

In order to detect the pruning parameters that lead to the best clustering result, an R routine was created which cycles the pruning method through a range of hmax, then gmin, then dmax using 0.1 increment steps. In each step, the quality of the clusters was assessed by calculation of the average Silhouette Coefficient (SC) and a cut-off of 0.51 was defined as the minimum required coefficient value for a reasonable structure to be found. The minimum number of members per cluster (N0) was set to 2, in order to make sure that true singletons that could not form a compact cluster core with sufficient separation from neighbouring clusters were left as outliers. The output of this routine returned the clustering parameters, the number of clusters and outliers, the average SC and an auxiliary index showing the ratio of outliers over clusters.

Multidimensional scaling was applied to all distance matrices and 2D maps were produced for visual inspection of the clusters. In addition, 3D maps were created and consulted through the visualisation tool GNUPLOT (Williams et al., 2007–2011), for better perception of the configuration of the global population of each CDR/length combination. The 2D/3D maps and the respective Silhouette Plots of pruning results with average SC greater than 0.51 and all positive individual Silhouette Widths (SW) were consulted in all cases in order to continually have a visual appreciation of the data configuration and clustering evolution, and to make informed decisions which allowed the final formalisation of the clustering procedure. Given that the desired clustering result would ideally produce as many well separated clusters and as few outliers as possible, the auxiliary index offered a quick composite comparison between pruning results, and was defined as: (2) a=1+S/C

where S is the number of outliers/singletons and C the number of clusters. The unit (1) was added to the index’s numerator in order to allow the comparison between pruning results with 0 outliers/singletons, but a different number of clusters.

Another index employed during the clustering procedure was that of the ideal maximum cluster diameter, which took into consideration the examined CDR length (l): (3) Di=1+l−910.

The rationale behind this formula was to define an ideal maximum diameter by adding or subtracting 0.1 Å per residue respectively above or below a length of 9. For a CDR with 9-residues, this diameter was set empirically at 1.0 Å, based on experience of manual 3D superpositions of CDR-L3/9-residues with the graphics program Swiss-PdbViewer (Spdbv; Guex & Peitsch, 1997). Observations suggested 1.0 Å to be an appropriate cut-off for significant visual conformational similarity for CDRs of this length. This auxiliary index played no further analytical role than to merely define a cut-off at which the possibility of cluster splitting was to be explored during the clustering procedure. In no case did it impose a diameter threshold for cluster formation. Conversely, cluster merging was explored between clusters that contained one or more members with greater affinity for the second cluster (revealed by its negative SW). If the merge resulted in a global average SC ≥ 0.51 then it was retained, otherwise the entire partition was discarded. In the end, the preferred clustering parameters were those that resulted in global average SC equal or higher than 0.51, all positive individual SWs and the lower auxiliary index α (Eq. (2)). If the number of outliers remained high, the optional PAM-stage was applied at the end of the tree cut procedure, but its results were only retained if all of the above partition quality criteria were satisfied.

When the optimal clustering result was obtained, the clusters’ cores, medoids, most distant members and their diameters were extracted for that CDR/length combination via a dedicated R routine. Clustering summaries were created with Java code, as well as lists and various post-analytical data that are detailed later.

Results

Clustering results

Tables of results were constructed for 58 CDR/length combination, gathering information that describes each individual cluster, which can be consulted for quick reference (Tables 3–7 for CDR-L1/-L2/-L3/-H1/-H2 and a separate supplementary table for CDR-H3, Supplemental Information 4). A summary table with all clustered lengths is available in Table 8. Detailed membership assignments can be found in two forms: one where every CDR is shown in alphabetical PDB order with all available clustering and data-mined information (cis/trans peptides, structure resolution, crystal space group, sequence, Ramachandran logos, cluster core label, bound state, light isotype, heavy or light chain only) and one where the same information is given in cluster order (Supplemental Information 6 and Supplemental Information 5). The ω-angle cut-off for cis-peptide detection was set to ± 30°; absence of cis-content that satisfied these limits resulted in an all-trans (allT) label. Bound state was flagged based on a list of bound antibodies obtained from SAbDab (Dunbar et al., 2014). This list did not contain idiotype-anti-idiotype complexes, therefore the 5 such files in the dataset were additionally flagged as bound (entries 1CIC, 1DVF, 1IAI, 1PG7, 3BQU).

Table 3 Summary table for the clustering of CDR-L1.

Unique sequence percentages per cluster are based on the total number of unique sequences in that length (i.e., total non-redundant sequences in that length, including outliers/singletons) and are rounded to the closest integer or to the closest first decimal, if lower than 0.5%. Two different clusters of the same length may contain the same CDR sequence.

Cluster	Population	Unique
sequences	Medoid
PDB
entry	Medoid
Ramachandran
conformation	Level-3
clusters	Level-2
clusters	Cluster diameter (Å)
(furthest members)	Best
resolution
in cluster (Å)
(PDB)	Species in cluster	Isotype	
CDR-L1 7-residues, total population: 2, total unique sequences: 1, clusters: 1, outliers/singletons: 0, Av.Silhouette: N/A	
L1-7-I	2	1 (100%)	3RPI_L	BBBGPBB	1	1	0.00 (3RPI_L-3RPI_B)	2.65 (3RPI_L)	HUMAN	κ	
CDR-L1 9-residues, total population: 10, total unique sequences: 4, clusters: 2, outliers/singletons: 0, Av.Silhouette: 0, 69	
L1-9-I	7	3 (75%)	3NGB_K	PBAPBBPPB	6	2	0.96 (3NGB_L-3Sé_L)	2.0 (3Sé_L)	HUMAN	κ	
L1-9-II	3	1 (25%)	3TV3_L	PLPDBAPBB	3	2	0.47 (3TYG_L-3TWC_L)	1.29 (3TV3_L)	HUMAN	λ	
CDR-L1 10-residues, total population: 127, total unique sequences: 28, clusters: 1, outliers/singletons: 1, Av.Silhouette: 0, 52	
L1-10-I	126	27 (96%)	1sy6_L	BPABPBABBB	31	3	0.91 (3C09_B-2Z92_B)	1.6 (3OZ9_L)	HUMAN, MOUSE	κ	
CDR-L1 11-residues, total population: 1042, total unique sequences: 180, clusters: 4, outliers/singletons: 9, Av.Silhouette: 0, 74	
L1-11-I	973	160 (89%)	2fjf_W	BPABPDGDPBB	120	28	1.29 (3fct_C-1ty7_L)	1.2 (3D9A_L)	HUMAN, MOUSE,
CHIMPANZE,
RABBIT, RAT,
SYNTHETIC
HUMANIZED
ANTIBODY, CHIMERA OF
MOUSE/HUMAN	κ	
L1-11-II	35	9 (5%)	1w72_L	PBPLAAABBPB	17	5	1.05 (2g75_D-1lil_A)	1.9 (3Q6G_L)	HUMAN,
HAMSTER	λ	
L1-11-III	23	7 (4%)	3MLV_L	PBADAADBPBB	7	1	0.76 (3UJI_L-1nfd_G)	1.6 (3UJI_L)	HUMAN, MOUSE	λ	
L1-11-IV	2	1 (1%)	1tzh_A	PBPBPAAPBBB	2	1	0.23 (1tzh_A-1tzh_L)	2.6 (1tzh_A)	MOUSE	κ	
CDR-L1 12-residues, total population: 82, total unique sequences: 26, clusters: 4, outliers/singletons: 1, Av.Silhouette: 0, 73	
L1-12-I	33	15 (58%)	1hq4_A	BBABPBPAADBB	13	5	1.02 (3LS4_L-3EO1_D)	1.9 (1ORS_A)	HUMAN, MOUSE	κ	
L1-12-II	24	6 (23%)	2fx7_L	BPABPPPLLPBB	7	1	0.66 (2b4c_L-1dn0_A)	1.76 (2fx7_L)	HUMAN	κ	
L1-12-III	14	2 (8%)	3JUY_C	BPABPBAALPBB	8	2	0.51 (1ob1_A-1n0x_M)	1.8 (1n0x_L)	HUMAN, MOUSE	κ	
L1-12-IV	10	2 (8%)	2OTU_A	BPPAADADPPBB	3	1	0.27 (2OTU_C-2GSG_C)	1.68 (2OTU_A)	MOUSE	λ	
CDR-L1 13-residues, total population: 81, total unique sequences: 26, clusters: 3, outliers/singletons: 0, Av.Silhouette: 0, 55	
L1-13-I	61	19 (73%)	3IYW_L	BBBAADAADBPBB	26	7	0.87 (2ig2_L-2b0s_L)	1.43 (3N9G_L)	HUMAN, MOUSE	λ	
L1-13-II	14	6 (23%)	1pew_B	BPABGPAAABPBB	5	1	0.66 (3H0T_A-3BDX_B)	1.6 (1pew_A)	HUMAN	λ	
L1-13-III	6	1 (4%)	3FKU_X	BBBAADAAAAGBB	6	2	0.35 (3FKU_U-3FKU_Y)	3.2 (3FKU_X)	HUMAN	λ	
CDR-L1 14-residues, total population: 207, total unique sequences: 25, clusters: 7, outliers/singletons: 14, Av.Silhouette: 0, 76	
L1-14-I	112	10 (40%)	1oar_L	BBAAGPPBAAALPB	36	5	1.08 (1oau_O-1nj9_L)	1.5 (1oaq_L)	HUMAN, MOUSE,
RAT, SYNTHETIC
HUMANIZED
ANTIBODY	λ	
L1-14-II	66	9 (36%)	3U4E_B	BBBAADAAABABBB	50	25	1.34 (1mcw_W-1mcq_A)	1.5 (3KDM_L)	HUMAN, SEAL	λ	
L1-14-III	3	1 (4%)	1lgv_B	BLAAAPPLAGDPBB	2	1	0.15 (1lhz_B-1jvk_B)	1.94 (1jvk_B)	HUMAN	λ	
L1-14-IV	3	1 (4%)	1jvk_A	BLAPPAPGBPDPBB	3	3	0.44 (1lhz_A-1lgv_A)	1.94 (1jvk_A)	HUMAN	λ	
L1-14-V	3	3 (12%)	7fab_L	BBBAADAADLBPBB	3	1	0.36 (3H42_L-1aqk_L)	1.84 (1aqk_L)	HUMAN	λ	
L1-14-VI	3	1 (4%)	2H3N_A	PPPGABPAADBPBB	3	2	0.39 (2H3N_C-2H32_A)	2.3 (2H3N_A)	HUMAN	λ	
L1-14-VII	3	1 (4%)	1mcs_B	PPAPPDPLPBDAPB	3	3	1.18 (1mcn_B-1mcc_B)	2.7 (1mcc_B)	HUMAN	λ	
CDR-L1 15-residues, total population: 80, total unique sequences: 34, clusters: 2, outliers/singletons: 48, Av.Silhouette: 0, 54	
L1-15-I	26	13 (38%)	2Y5T_B	BBABPDPBLLBPPBB	14	3	0.67 (3ZTJ_L-2nz9_C)	2.0 (1h0d_A)	HUMAN, MOUSE	κ	
L1-15-II	6	3 (9%)	1i9r_L	BPABPDBBADBBPBB	4	1	0.61 (3PHO_A-1i7z_C)	2.0 (3PHQ_A)	HUMAN, MOUSE	κ	
CDR-L1 16-residues, total population: 352, total unique sequences: 74, clusters: 5, outliers/singletons: 33, Av.Silhouette: 0, 60	
L1-16-I	309	66 (89%)	3QCU_L	BBABPAPPAALPBPBB	81	13	1.24(2GJZ_A-1f3d_L)	1.22 (1mju_L)	HUMAN, MOUSE,
CHIMERA OF
MOUSE/HUMAN	κ	
L1-16-II	3	1 (1%)	1cfv_L	BPABPDDABABPLPBB	3	1	0.22 (2bfv_L-1bfv_L)	2.1 (1bfv_L)	MOUSE	κ	
L1-16-III	3	1 (1%)	3FO9_L	BPABPDPABPLBBPBB	3	1	0.36 (3FO9_A-1axt_L)	1.9 (3FO9_L)	MOUSE	κ	
L1-16-IV	2	1 (1%)	1l7s_L	BPABPPPPBLGPPABB	1	1	0.00 (1VPO_L-1l7s_L)	2.15 (1l7s_L)	MOUSE	κ	
L1-16-V	2	1 (1%)	1nak_L	BBABPDGBDLDDPPBB	1	1	0.04 (1nak_M-1nak_L)	2.57 (1nak_L)	MOUSE	κ	
CDR-L1 17-residues, total population: 172, total unique sequences: 36, clusters: 1, outliers/singletons: 1, Av.Silhouette: N/A	
L1-17-I	171	36 (100%)	3MNZ_A	BBABPDPPAADLPPPBB	71	17	1.51 (1xcq_G-1him_H)	1.45 (1q9r_A)	HUMAN, MOUSE,
CHIMERA OF
MOUSE/HUMAN	κ	
Total
(level-1
clusters
only)	2048				533	140					

Table 4 Summary table for the clustering of CDR-L2.

Cluster	Population	Unique
sequences	Medoid
PDB
entry	Medoid
Ramachandran
conformation	Level-3
clusters	Level-2
clusters	Cluster diameter (Å)
(furthest members)	Best resolution
in cluster (Å)
(PDB)	Species in cluster	Isotype	
CDR-L2 7-residues, total population: 2161, total unique sequences: 278, clusters: 3, outliers/singletons: 2, Av.Silhouette: 0, 61	
L2-7-I	2109	272 (98%)	1dn0_C	LLDPPPP	121	45	1.56 (3KYM_O-1nj9_A)	1.2 (3D9A_L)	HUMAN, MOUSE,
RABBIT, RAT,
SEAL, HAMSTER,
CHIMPANZEE, RAT,
SYNTHETIC
HUMANIZED
ANTIBODY, CHIMERA OF
MOUSE/HUMAN	κ, λ	
L2-7-II	39	8 (3%)	3RIA_L	GADBBPP	12	5	1.04 (3RHW_K-3FKU_X)	1.66 (3QHF_L)	HUMAN, MOUSE	κ, λ	
L2-7-III	11	2 (1%)	2a6k_L	LLGGPPD	8	3	0.75 (2ZJS_L-2V7H_L)	2.5 (2a6i_A)	MOUSE	κ	
CDR-L2 11-residues, total population: 13, total unique sequences: 3, clusters: 2, outliers/singletons: 0, Av.Silhouette: 0, 90	
L2-11-I	10	2 (67%)	2OTU_A	BPAALPBBPPP	1	1	0.39 (2GSG_C-2OTU_C)	1.68 (2OTU_A)	MOUSE	λ	
L2-11-II	3	1 (33%)	2H32_A	BDBAABBBPPA	3	2	0.2 (2H3N_C-2H3N_A)	2.3 (2H3N_A)	HUMAN	λ	
Total (level-1 clusters only)	2172				145	56					

Table 5 Summary table for the clustering of CDR-L3.

Cluster	Population	Unique
sequences	Medoid
PDB
entry	Medoid
Ramachandran
conformation	Level-3
clusters	Level-2
clusters	Cluster diameter (Å)
(furthest members)	Best resolution
in cluster (Å)
(PDB)	Species in cluster	Isotype	
CDR-L3 5-residues, total population: 10, total unique sequences: 4, clusters: 1, outliers/singletons: 0, Av.Silhouette: N/A	
L3-5-I	10	4 (100%)	3NGB_C	BBGAB	2	1	0.26 (3U7W_L-3Sé_L)	1.9 (3SE8_L)	HUMAN	κ	
CDR-L3 7-residues, total population: 5, total unique sequences: 2, clusters: 1, outliers/singletons: 0, Av.Silhouette: N/A	
L3-7-I	5	2 (100%)	3IU3_L	BBDDDLP	4	1	0.49 (1mim_L-1dfb_L)	2.6 (1mim_L)	HUMAN, MOUSE	κ	
CDR-L3 8-residues, total population: 138, total unique sequences: 43, clusters: 6, outliers/singletons: 2, Av.Silhouette: 0, 56	
L3-8-I	105	28 (65%)	1q9w_C	BPDABGBB	13	4	0.76 (3KJ4_L-1m71_A)	1.45 (1q9r_A)	HUMAN, MOUSE,
RAT	κ	
L3-8-II	12	6 (14%)	1tzh_L	BPDBBPBP	9	5	1.18 (3DGG_C-1e6j_L)	1.77 (2fat_L)	HUMAN, MOUSE	κ	
L3-8-III	7	4 (9%)	3hfl_L	BPDPABPB	7	4	0.98 (1ORS_A-1ehl_L)	1.7 (1YQV_L)	MOUSE	κ	
L3-8-IV	5	2 (5%)	3JWD_L	BPDADPDP	3	3	0.66 (2J88_L-1rz7_L)	2.0 (1rz7_L)	HUMAN, MOUSE	κ	
L3-8-V	4	3 (7%)	1za3_A	BPAABPDP	4	4	0.85 (3DGG_A-1za3_L)	2.3 (3DGG_A)	HUMAN	κ	
L3-8-VI	3	1 (2%)	3MCL_L	BBAAPBPA	1	1	0.24 (3O11_A-3O11_L)	1.7 (3MCL_L)	CHIMERA OF
MOUSE/HUMAN	κ	
CDR-L3 9-residues, total population: 1725, total unique sequences: 358, clusters: 6, outliers/singletons: 5, Av.Silhouette: 0, 65	
L3-9-I	1528	328 (92%)	1dqj_A	BBDABPPPB	153	34	1.72 (1maj_A_11-1kel_L)	1.2 (3D9A_L)	HUMAN, MOUSE,
RAT, RABBIT,
CHIMPANZEE,
SYNTHETIC
HUMANIZED
ANTIBODY,
CHIMERA OF
MOUSE/HUMAN	κ	
L3-9-II	136	20 (6%)	2XZQ_L	BBBBGDBPB	36	8	1.36 (2E27_L-1pw3_B)	1.5 (1oaq_L)	HUMAN, MOUSE,
RAT,
SYNTHETIC
HUMANIZED
ANTIBODY,
CHIMERA OF
MOUSE/HUMAN	κ, λ	
L3-9-III	40	8 (2%)	1op3_L	BPBBADBBB	15	8	1.4 (8fab_C-3C08_L)	1.75 (1op3_L)	HUMAN, MOUSE,
CHIMERA OF
MOUSE/HUMAN	κ, λ	
L3-9-IV	8	1 (0,3%)	3MLR_L	BBBBBBBPA	3	2	1.24 (3MLV_M-3MLS_M)	1.8 (3MLR_L)	HUMAN	λ	
L3-9-V	6	1 (0,3%)	3RU8_L	BBBGLLBBB	2	2	0.34 (1n0x_L-1hzh_M)	1.8 (1n0x_L)	HUMAN	κ	
L3-9-VI	2	1 (0,3%)	3FN0_L	BBGBPBABA	1	1	0.23 (3Q1S_L-3FN0_L)	1.8 (3FN0_L)	HUMAN	κ	
CDR-L3 10-residues, total population: 113, total unique sequences: 27, clusters: 12, outliers/singletons: 6, Av.Silhouette: 0, 59	
L3-10-I	22	2 (7%)	3MUG_I	BBPBADLPBB	4	2	0.76 (3MUH_L-3LRS_B)	1.8 (3U2S_L)	HUMAN	λ	
L3-10-II	20	2 (7%)	1MCD_A	BBPBGLLBBP	15	9	1.28 (1mcc_A-1lil_B)	2.0 (2mcg_2)	HUMAN, SEAL	λ	
L3-10-III	9	4 (15%)	3B5G_A	BBBPAALPPB	4	1	1.19 (3GO1_L-2XZA_L)	1.36 (2XZC_L)	HUMAN	λ	
L3-10-IV	9	4 (15%)	3EYO_C	BBDABBPPPB	5	2	0.51 (3F12_A-3eyq_C)	1.8 (1jgu_L)	MOUSE	κ	
L3-10-V	7	3 (11%)	2dd8_L	BBBBADDGPB	5	1	1.06 (3UJI_L-3G6A_L)	1.6 (3UJI_L)	HUMAN	λ	
L3-10-VI	5	1 (4%)	3MLY_L	BBPBAALBPB	1	1	0.24 (3MLZ_L-3MLY_M)	1.7 (3MLY_L)	HUMAN	λ	
L3-10-VII	5	3 (11%)	3TV3_L	BBPBGADPBB	4	3	0.84 (3TWC_L-1mcw_M)	1.29 (3TV3_L)	HUMAN	λ	
L3-10-VIII	4	1 (4%)	2fl5_A	BBBPADLPPB	2	2	0.18 (2fl5_C-2fl5_L)	3.0 (2fl5_L)	HUMAN	λ	
L3-10-IX	4	1 (4%)	1mqk_L	BPDBGBPPBB	3	1	0.31 (3HB3_D-1qle_L)	1.28 (1mqk_L)	MOUSE	κ	
L3-10-X	3	1 (4%)	3IDY_L	BBDABBPPBB	2	1	0.23 (3IDY_C-3IDX_L)	2.5 (3IDX_L)	HUMAN	κ	
L3-10-XI	2	1 (4%)	1i7z_A	BPBBABPPBB	2	1	0.12 (1i7z_A-1i7z_C)	2.3 (1i7z_A)	CHIMERA OF
MOUSE/HUMAN	κ	
L3-10-XII	17	3 (11%)	1mcn_B	BBPPPADBBP	17	15	1.55 (1mch_B-1lil_A)	2.0 (2mcg_1)	HUMAN, SEAL	λ	
CDR-L3 11-residues, total population: 142, total unique sequences: 38, clusters: 9, outliers/singletons: 7, Av.Silhouette: 0, 64	
L3-11-I	74	24 (63%)	3G04_A	BBPBAADLBPB	27	2	1.72 (3MAC_L-2rhe_A)	1.43 (3N9G_L)	HUMAN, MOUSE,
HAMSTER	κ, λ	
L3-11-II	24	1 (3%)	1yym_Q	BPDAPBPPBPB	6	3	0.34 (1yym_L-1g9n_L)	1.99 (2NY1_C)	HUMAN	κ	
L3-11-III	11	3 (8%)	2OMN_A	BBPBAPABABB	7	3	1.01 (3GHE_L-2OMB_D)	1.5 (3KDM_L)	HUMAN	λ	
L3-11-IV	8	1 (3%)	2QR0_G	BBBLPDDBABB	8	4	0.36 (2QR0_S-2QR0_K)	3.5 (2QR0_A)	HUMAN	κ	
L3-11-V	5	2 (5%)	3NH7_M	BBPBAPLLBBB	4	3	0.59 (4bjl_A-3NH7_O)	2.4 (4bjl_A)	HUMAN	λ	
L3-11-VI	4	2 (5%)	2JB6_A	BBBPAALDBBB	2	1	0.77 (3UJJ_L-2JB5_L)	2.0 (3UJJ_L)	HUMAN	λ	
L3-11-VII	4	1 (3%)	3EFF_C	BBDAPBBLAGB	4	4	0.6 (3PJS_C-3EFF_A)	3.8 (3EFF_A)	MOUSE	κ	
L3-11-VIII	3	1 (3%)	2b1h_L	BBPBDAALBPB	2	1	0.2 (2b1a_L-2b0s_L)	2.0 (2b1h_L)	HUMAN	λ	
L3-11-IX	2	1 (3%)	1nfd_E	BBBBGAALPPB	1	1	0.17 (1nfd_E-1nfd_G)	2.8 (1nfd_E)	MOUSE	λ	
CDR-L3 12-residues, total population: 19, total unique sequences: 6, clusters: 4, outliers/singletons: 0, Av.Silhouette: 0, 73	
L3-12-I	6	1 (17%)	2X7L_D	BBGBAAGADGBB	1	1	0.04 (2X7L_K-2X7L_B)	3.17 (2X7L_B)	Not available	κ	
L3-12-II	6	1 (17%)	1q1j_L	BBBPAPAALPPB	2	1	0.31 (3GHB_L-3C2A_M)	2.1 (3C2A_L)	HUMAN	λ	
L3-12-III	4	2 (33%)	3LZF_L	BBPBAPGAGBPB	3	3	0.9 (3QHZ_M-3GBN_L)	1.55 (3QHZ_M)	HUMAN	λ	
L3-12-IV	3	2 (33%)	3LQA_L	BBPBAPGAAGPB	3	3	1.35 (3P30_L-3LMJ_L)	2.2 (3LMJ_L)	HUMAN	λ	
CDR-L3 13-residues, total population: 12, total unique sequences: 2, clusters: 3, outliers/singletons: 1, Av.Silhouette: 0, 71	
L3-13-I	6	1 (50%)	2OTU_A	BBBBPPAABPBBB	1	1	0.18 (2OTW_C-2OTU_G)	1.68 (2OTU_A)	MOUSE	λ	
L3-13-II	3	1 (50%)	2QHR_L	BBBBBBLLBPBBB	3	3	0.97 (3FFD_B-2GSG_A)	2.0 (2QHR_L)	MOUSE	λ	
L3-13-III	2	1 (50%)	3MLW_L	BBBPDABABPPPB	2	2	0.48 (3MLW_L-3MLW_M)	2.7 (3MLW_L)	HUMAN	λ	
Total
(level-1
clusters
only)	2143				393	153					

Table 6 Summary table for the clustering of CDR-H1.

Level-2 clusters are shown exceptionally (marked with an asterisk) when no level-1 cluster is formed (minimum of 2 members required).

Cluster	Population	Unique
sequences	Medoid
PDB
entry	Medoid
Ramachandran
conformation	Level-3
clusters	Level-2
clusters	Cluster diameter (Å)
(furthest members)	Best resolution
in cluster (Å)
(PDB)	Species in cluster	
CDR-H1 10-residues, total population: 6, total unique sequences: 2, clusters: 1, outliers/singletons: 0, Av.Silhouette: N/A	
H1-10-I	6	2 (100%)	1kxq_H	BPABPBABBB	2	1	0.61 (3eba_A-1kxq_F)	1.6 (1kxq_E)	CAMEL, HUMAN	
CDR-H1 12-residues, total population: 2, total unique sequences: 2, clusters: 0, outliers/singletons: 2, Av.Silhouette: N/A	
H1-12-O-1*	1	1 (50%)	1ghf_H	BBBBPAAABPBB	1	1	N/A	2.7 (1ghf_H)	MOUSE	
H1-12-O-2*	1	1 (50%)	3IY2_B	PBBLBABBABBB	1	1	N/A	18.0 (3IY2_B)	MOUSE	
CDR-H1 13-residues, total population: 1845, total unique sequences: 450, clusters: 11, outliers/singletons: 164, Av.Silhouette: 0, 54	
H1-13-I	1555	390 (87%)	2OSL_A	PPBLBPAADBPBB	214	23	1.74 (2GK0_B-1rzi_J)	1.2 (3D9A_H)	MOUSE, HUMAN,
RAT, RABBIT,
LLAMA, HAMSTER,
CHIMPANZEE,
CHIMERA OF
MOUSE/HUMAN,
SYNTHETIC
HUMANIZED
ANTIBODY	
H1-13-II	49	6 (1%)	3F7Y_A	PBBGPBBAAPBBB	19	6	1.77 (3QXW_D-3GKZ_A)	1.72 (2IH3_A)	MOUSE, HUMAN,
RAT, LLAMA,
CHIMERA OF
MOUSE/HUMAN	
H1-13-III	24	4 (1%)	1bzq_K	BPBLPABBPABBB	11	3	1.12 (2P42_D-1jtp_B)	1.1 (2P45_B)	MOUSE,
DROMEDARY,
CAMEL	
H1-13-IV	8	2 (0.4%)	3QXV_A	BBABPBAPPBPBB	7	2	0.97 (3QXV_B-1YC7_A)	1.6 (1YC7_A)	DROMEDARY,
CAMEL	
H1-13-V	9	3 (1%)	2X7L_J	PPBLPAPABBPBB	4	2	1.11 (2W9E_H-1YC8_B)	2.7 (1YC8_B)	CAMEL	
H1-13-VI	9	1 (0.2%)	2WZP_E	BPABBABPLPBBB	4	2	0.35 (2WZP_J-2BSE_E)	2.6 (2WZP_D)	CAMELID	
H1-13-VII	2	1 (0.2%)	1SHM_A	PPBGPAAPPBPBB	2	1	0.19 (1SHM_A-1SHM_B)	1.9 (1SHM_A)	HUMAN, MOUSE,
LLAMA	
H1-13-VIII	8	5 (1%)	3EZJ_B	BPBGPAAAPDBBB	6	2	1.05 (1zvy_A-1SJX_A)	1.5 (1zvh_A)	HUMAN, CAMEL, LLAMA	
H1-13-IX	3	1 (0.2%)	3GBM_H	BBPGGAPBDBPBB	3	1	0.17 (3GBN_H-3GBM_I)	2.2 (3GBN_H)	HUMAN	
H1-13-X	3	1 (0.2%)	2X89_A	BBBLPPLLBBPBB	2	1	0.22 (2X89_C-2X89_B)	2.16 (2X89_A)	HUMAN	
H1-13-XI	2	1 (0.2%)	1ngx_B	PPALPPBABBPBB	1	1	0.01 (1ngx_H-1ngx_B)	1.8 (1ngx_B)	HUMAN, MOUSE	
CDR-H1 14-residues, total population: 72, total unique sequences: 17, clusters: 1, outliers/singletons: 2, Av.Silhouette: 0, 64	
H1-14-I	70	16 (94%)	1kcv_H	BBBLBPAAABGBBB	47	11	1.37 (2f58_H-2ajz_H)	1.3 (1ncw_H)	HUMAN, MOUSE	
CDR-H1 15-residues, total population: 128, total unique sequences: 29, clusters: 3, outliers/singletons: 3, Av.Silhouette: 0, 62	
H1-15-I	117	25 (86%)	2HWZ_H	BBBLBBAAPPLPBBB	43	12	1.68 (3BQU_B-3B2V_H)	1.5 (3IFL_H)	HUMAN, MOUSE	
H1-15-II	6	2 (7%)	3BAE_H	BBBLBBAAAALPPBB	4	2	0.86 (3BKC_H-3AAZ_A)	1.59 (3BAE_H)	HUMAN, MOUSE	
H1-15-III	2	1 (3%)	3FZU_H	BBBLPAPPAADBPBB	2	1	0.28 (3FZU_C-3FZU_H)	2.5 (3FZU_H)	HUMAN	
CDR-H1 16-residues, total population: 3, total unique sequences: 2, clusters: 1, outliers/singletons: 1, Av.Silhouette: N/A	
H1-16-I	2	1 (50%)	3eak_B	PBBGLAABPAAAPPBB	1	1	0.42 (3eak_A-3eak_B)	1.95 (3eak_A)	CAMEL	
CDR-H1 24-residues, total population: 1, total unique sequences: 1, clusters: 0, outliers/singletons: 1, Av.Silhouette: N/A	
H1-24-O-1*	1	1 (100%)	3K3Q_A	PPBLBALDLGAAG
AADAADBGBBB	1	1	N/A	2.6 (3K3Q_A)	LLAMA	
Total
(level-1
clusters
only)	1884				379	77				

Table 7 Summary table for the clustering of CDR-H2.

Level-2 clusters are shown exceptionally (marked with an asterisk) when no level-1 cluster is formed (minimum of 2 members required).

Cluster	Population	Unique
sequences	Medoid
PDB
entry	Medoid
Ramachandran
conformation	Level-3
clusters	Level-2
clusters	Cluster diameter (Å)
(furthest members)	Best resolution
in cluster (Å)
(PDB)	Species in cluster	
CDR-H2 8-residues, total population: 6, total unique sequences: 2, clusters: 1, outliers/singletons: 0, Av.Silhouette: N/A	
H2-8-I	6	2 (100%)	1f2x_K	BBBGAPBB	3	3	1.00 (2OJZ_H-1f2x_K)	1.89 (2OK0_H)	CAMEL, MOUSE	
CDR-H2 9-residues, total population: 436, total unique sequences: 117, clusters: 6, outliers/singletons: 1, Av.Silhouette: 0, 68	
H2-9-I	412	110 (94%)	1fe8_I	BBPAALPBB	31	9	1.47 (3K81_B-2aj3_B)	1.2 (3D9A_H)	HUMAN, MOUSE,
RAT, CAMEL,
LLAMA,
CHIMERA OF
MOUSE/HUMAN,
HAMSTER, ALPACA,
RABBIT	
H2-9-II	11	3 (3%)	2ak1_H	BBBLLDBBB	3	2	1.03 (2aju_H-1mco_H)	1.5 (2aju_H)	HUMAN, MOUSE	
H2-9-III	4	1 (1%)	3UAJ_H	BBBADPPPB	2	1	0.14 (3UC0_H-3UAJ_H)	2.71 (3UC0_H)	CHIMPANZEE	
H2-9-IV	4	2 (2%)	1ken_H	BLPAAGAAG	4	4	1.46 (1ken_T-1bgx_H)	2.2 (1ay1_H)	MOUSE	
H2-9-V	2	1 (1%)	1YC7_A	BBBPPLPBB	1	1	0.13 (1YC7_A-1YC7_B)	1.6 (1YC7_A)	CAMEL	
H2-9-VI	2	1 (1%)	2aj3_D	BBBABGPBB	1	1	0.19 (2aj3_D-2aj3_F)	2.03 (2aj3_D)	HUMAN	
CDR-H2 10-residues, total population: 1508, total unique sequences: 381, clusters: 10, outliers/singletons: 152, Av.Silhouette: 0, 56	
H2-10-I	822	238 (62%)	1uyw_H	BBPAAALPBB	62	16	1.40 (1rzi_J-1a5f_H)	1.22 (1mju_H)	HUMAN, MOUSE,
LLAMA, CAMEL,
RAT, SYNTHETIC
HUMANIZED
ANTIBODY	
H2-10-II	417	101 (27%)	1i8i_B	BBPAALABBB	48	18	1.46 (3GJE_H-1R24_B)	1.06 (2X1Q_A)	HUMAN, MOUSE,
LLAMA, CAMEL	
H2-10-III	75	1 (0.3%)	2KH2_B_20	BBPAALDPBP	35	6	1.31
(2KH2_B_50-2KH2_B_17)	N/A	MOUSE	
H2-10-IV	19	1 (0.3%)	1vhp_A_11	BBPGALAPBB	3	2	0.88 (1vhp_A_5-1vhp_A_6)	N/A	HUMAN	
H2-10-V	7	2 (1%)	1G9E_A	BBPBDLDBPB	7	3	1.19 (1G9E_9-1d6v_H)	2.0 (1d6v_H)	LLAMA,
CHIMERA OF
MOUSE/HUMAN	
H2-10-VI	4	1 (0.3%)	1bzq_M	BBPAPABBPB	1	1	0.03 (1bzq_L-1bzq_N)	2.8 (1bzq_K)	DROMEDARY	
H2-10-VII	4	2 (1%)	1cfv_H	BBPAALBPDB	2	2	0.86 (1zv5_A-1bfv_H)	2.0 (1zv5_A)	MOUSE, CAMEL	
H2-10-VIII	3	1 (0.3%)	2fd6_H	BBBGBAABBB	2	2	0.89 (3BT2_H-2fat_H)	1.77 (2fat_H)	MOUSE	
H2-10-IX	3	2 (1%)	2fjg_B	BPBAPLLPPB	3	2	0.97 (3LMJ_H-2fjg_H)	2.2 (3LMJ_H)	HUMAN	
H2-10-X	2	1 (0.3%)	3NCY_Q	BBPAADPBBB	1	1	0.00 (3NCY_Q-3NCY_P)	3.2 (3NCY_Q)	MOUSE	
CDR-H2 11-residues, total population: 3, total unique sequences: 3, clusters: 0, outliers/singletons: 3, Av.Silhouette: N/A	
H2-11-O-1-1*	1	1 (33%)	2X6M_A	BBPPLLPDPBB	1	1	N/A	1.62 (2X6M_A)	DROMEDARY	
H2-11-O-2-1*	1	1 (33%)	3SE9_H	BBPAADLPBBB	1	1	N/A	2.0 (3SE9_H)	HUMAN	
H2-11-O-3-1*	1	1 (33%)	3H0T_B	BBBBBLBPBBB	1	1	N/A	1.89 (3H0T_B)	HUMAN	
CDR-H2 12-residues, total population: 171, total unique sequences: 38, clusters: 4, outliers/singletons: 0, Av.Silhouette: 0, 78	
H2-12-I	160	36 (95%)	3IJH_B	BBPPAAALLPBB	19	7	1.62 (4fab_H-2aeq_H)	1.45 (1dlf_H)	HUMAN, MOUSE,
RAT, CHIMERA OF
MOUSE/HUMAN	
H2-12-II	4	1 (3%)	1aif_H	BPBDALPABBBB	2	2	1.32 (1iai_I-1aif_H)	2.9 (1aif_H)	MOUSE	
H2-12-III	4	1 (3%)	3IXX_G	Not available	1	1	0.1 (3IXX_I-3IXY_G)	15.0 (3IXX_G)	MOUSE	
H2-12-IV	3	1 (3%)	3QHZ_I	BBPAAPAPBBBP	2	2	0.4 (3QHZ_H-3LZF_H)	1.55 (3QHZ_H)	HUMAN	
CDR-H2 15-residues, total population: 6, total unique sequences: 3, clusters: 2, outliers/singletons: 1, Av.Silhouette: 0, 84	
H2-15-I	3	1 (33%)	1i3v_B	BBPDBPABADBPP	3	1	0.55 (1i3v_A-1i3u_A)	1.95 (1i3u_A)	LLAMA	
H2-15-II	2	1 (33%)	3TYG_H	BBBAPBBADBDGBBB	2	1	0.29 (3TYG_H-3TV3_H)	1.29 (3TV3_H)	HUMAN	
Total
(level-1
clusters
only)	1973				238	88				

Comparison of clustering results

The level-1 clusters obtained in this work were compared to the clustering results of previous major CDR studies (Tables 9–13 for CDR-L1, -L2, -L3, -H1 and -H2, Supplemental Information 2 for CDR-H3). Specifically, comparisons were made with the clusters found in Martin & Thornton (1996) because it was the first five CDR clustering performed on a significant CDR dataset (57 antibody structures, 269 CDRs), presented most major conformational classes and for these reasons is regularly cited in research of this kind. Comparisons were also made with the clustering results in North, Lehmann & Dunbrack (2011) as this is the most recent relevant analysis, which used the largest CDR dataset (932 antibody structures before filtering, 1897 CDRs after filtering) until the present study. Also included were the results from Kuroda et al. (2009) for the comparisons in CDR-L3, as this recent dedicated analysis used an RMSD-based approach, as is the case in this work, while using a considerable number of CDR structures (212 CDR-L3 structures). For the first five CDRs, the present study comprised 1,359 antibody structures and 10,680 CDRs (and a total of 12,712 CDRs including CDR-H3). Commenting on these comparisons is made in the discussion section below.

Table 8 Summation of clustered lengths per CDR.

(A) Summation of clustered lengths per CDR, with population, non-redundant sequences, number of clusters and outliers information. CDR lengths that were clustered for the first time are highlighted in bold/italics. (B) The complete CDR-H3 conformation, using the H95-H102 extents definition, has not been extensively clustered before; therefore only lengths that were not considered in Kuroda et al. (2009) are noted as new for conformity with the literature. CDR-H3 lengths 4 and 24 are marked with an asterisk as the corresponding structures are also found in North, Lehmann & Dunbrack (2011), but acknowledged as 2 residues longer, due to different CDR-H3 extents (H93-H102).

(A)	
CDR	Observed lengths
(new lengths)	Total structure
population	Unique
sequences	Level-1
clusters	Level-1 only
structure population	Singletons/outliers	
L1	7	2	1	1	2	0	
9	10	4	2	10	0	
10	127	28	1	126	1	
11	1,042	180	4	1,033	9	
12	82	26	4	81	1	
13	81	26	3	81	0	
14	207	25	7	193	14	
15	80	34	2	32	48	
16	352	74	5	319	33	
17	172	36	1	171	1	
Total	10 lengths	2,155	434	30	2,048	107	
CDR	Observed lengths	Total structure
population	Unique
sequences	Level-1
clusters	Level-1 only
structure population	Singletons/outliers	
L2	7	2,161	278	3	2,159	2	
11	13	3	2	13	0	
Total	2 lengths	2,174	281	5	2,172	2	
CDR	Observed lengths
(new lengths)	Total structure
population	Unique
sequences	Level-1
clusters	Level-1 only
structure population	Singletons/outliers	
L3	5	10	4	1	10	0	
7	5	2	1	5	0	
8	138	43	6	136	2	
9	1,725	358	6	1,720	5	
10	113	27	12	107	6	
11	142	38	9	135	7	
12	19	6	4	19	0	
13	12	2	3	11	1	
Total	8 lengths	2,164	480	42	2,143	21	
CDR	Observed lengths
(new lengths)	Total structure
population	Unique
sequences	Level-1
clusters	Level-1 only
structure population	Singletons/outliers	
H1	10	6	2	1	6	0	
12	2	2	0	0	2	
13	1,845	450	11	1,681	164	
14	72	17	1	70	2	
15	128	29	3	125	3	
16	3	2	1	2	1	
24	1	1	0	0	1	
Total	7 lengths	2,057	503	17	1,884	173	
(B)	
CDR	Observed lengths
(new lengths)	Structure
population	Unique
sequences	Level-1
clusters	Level-1 only
structure population	Singletons/outliers	
H2	8	6	2	1	6	0	
9	436	117	6	435	1	
10	1,508	381	10	1,356	152	
11	3	3	0	0	3	
12	171	38	4	171	0	
15	6	3	2	5	1	
Total	6 lengths	2,130	544	23	1,973	157	
CDR	Observed lengths
(new lengths)	Structure
population	Unique
sequences	Level-1
clusters	Level-1 only
structure population	Singletons/outliers	
H3	3	18	4	1	18	0	
4*	38	12	2	36	2	
5	93	28	6	85	8	
6	33	12	3	30	3	
7	97	41	7	69	28	
8	168	46	7	141	27	
9	181	55	8	132	49	
10	377	98	35	292	85	
11	231	64	26	151	80	
12	206	51	21	174	32	
13	130	42	22	105	25	
14	128	40	19	104	24	
15	96	23	18	81	15	
16	40	16	8	28	12	
17	28	14	6	19	9	
18	37	11	6	31	6	
19	48	12	9	46	2	
20	13	4	3	13	0	
21	10	1	1	10	0	
22	33	4	2	31	2	
23	1	1	0	0	1	
24*	12	2	2	12	0	
25	1	1	0	0	1	
28	12	2	1	12	0	
31	1	1	0	0	1	
Total	25 lengths	2,032	585	213	1,620	412	
Cumulative
total
(all CDRs)	58 lengths	12,712	2,827	330	11,840	872	

Table 9 Comparison of level-1 conformational clusters obtained in CDR-L1 with external sets.

The cluster medoid/median or representative of the external sets was used for identification of correspondences. In brackets, next to each correspondence, is the full, 3-level classification in this work of the representative of the external set and the number of corresponding members in full population comparison. Martin & Thornton (1996) cluster 14F is marked with a question mark, because its representative (2BJL, superseded by 4BJL) actually has a 13-residue CDR-L1.

This work
[CDR-L1 cluster]	Martin & Thornton, 1996
〈corresponding cluster/canonical〉
(level-3 of external median)
(corresponding members)	North, Lehmann & Dunbrack, 2011
〈corresponding cluster〉
(level-3 of external median)
(corresponding members)	
L1-7-I	–	–	
L1-9-I	–	–	
L1-9-II	–	–	
L1-10-I	10A/1 (L1-10-I-1-1) (4/4)	L1-10-1 (L1-10-I-1-1) (20/20)
L1-10-2 (L1-10-I-2-2) (2/2)	
L1-11-I	11A/2 (L1-11-I-2-1) (22/22)	L1-11-1 (L1-11-I-1-2) (76/76)
L1-11-2 (L1-11-I-2-1) (55/55)	
L1-11-II	–	L1-11-3 (L1-11-II-1-2) (3/5)	
L1-11-III	11B/- (L1-11-III-1-1) (1/1)	–	
L1-11-IV	–	–	
L1-12-I	–	L1-12-1 (L1-12-I-1-1) (5/5)	
L1-12-II	–	L1-12-2 (L1-12-II-1-2) (4/5)	
L1-12-III	–	–	
L1-12-IV	–	L1-12-3 (L1-12-IV-1-2) (2/2)	
L1-13-I	13A/5λ(L1-13-I-1-2) (2/2)
14F/-?(L1-13-I-7-1) (1/1)	L1-13-1 (L1-13-I-1-2) (7/7)	
L1-13-II	–	L1-13-2 (L1-13-II-1-1) (4/4)	
L1-13-III	–	–	
L1-14-I	14B/7λ(L1-14-I-2-3) (3/3)	L1-14-1 (L1-14-I-1-3) (14/14)	
L1-14-II	14C/- (L1-14-II-13-1) (1/1)
14E/-(L1-14-II-14-1) (1/1)	L1-14-2 (L1-14-II-4-1) (3/4)	
L1-14-III	–	–	
L1-14-IV	–	–	
L1-14-V	14A/6λ(L1-14-V-1-2) (1/1)	–	
L1-14-VI	–	–	
L1-14-VII	–	–	
L1-15-I	–	L1-15-1 (L1-15-I-1-11) (8/11)	
L1-15-II	–	–	
L1-16-I	16A/4 (L1-16-I-1-51) (8/9)
16C/-(L1-16-I-1-20) (1/1)	L1-16-1 (L1-16-I-1-1) (62/68)	
L1-16-II	–	–	
L1-16-III	–	–	
L1-16-IV	–	–	
L1-16-V	–	–	
L1-17-I	17A/3 (L1-17-I-1-17) (4/4)	L1-17-1 (L1-17-I-1-3) (21/21)	
Outliers			
L1-12-O	12A/6 (L1-12-O-1-1) (1/1)	–	
L1-14-O	14D/- (L1-14-O-3-1) (1/1)	–	
L1-15-O	15A/5 (L1-15-O-6-1) (1/1)
15B/- (L1-15-O-1-4) (2/2)	L1-15-2 (L1-15-O-3-1) (2/2)	
L1-16-O	16B/- (L1-16-O-8-1) (2/2)	–	

Table 10 Comparison of level-1 conformational clusters obtained in CDR-L2 with external sets.

See notes in Table 9. In North, Lehmann & Dunbrack (2011), the CDR extents were defined as L49-L56, instead of L50-L56; hence a direct comparison is not possible. Nonetheless, since position L49 is fairly conserved structurally and for reference reasons, a correspondence of the longer by 1 residue clusters is shown, based on the representative of those clusters (in square brackets and in full-italics).

This work
[CDR-L2 cluster]	Martin & Thornton, 1996
〈corresponding cluster/canonical〉
(level-3 of external median)
(corresponding members)	North, Lehmann & Dunbrack, 2011
〈corresponding cluster〉
(level-3 of external median)
(corresponding members)	
L2-7-I	7A/1 (L2-7-I-2-1) (55/55)	[L2-8-1 (L2-7-I-2-1) (290/290)
L2-8-2 (L2-7-I-6-2) (9/9)
L2-8-4 (L2-7-I-10-1) (2/2)
L2-8-5 (L2-7-I-14-2) (2/2)]	
L2-7-II	–	[L2-8-3 (L2-7-II-1-2) (3/3)]	
L2-7-III	7B/1 (L2-7-III-1-6) (1/1)	–	
L2-11-I	–	[L2-12-2 (L2-11-I-1-1) (2/2)]	
L2-11-II	–	[L2-12-1 (L2-11-II-2-1) (2/2)]	

Table 11 Comparison of level-1 conformational clusters obtained in CDR-L3, with external sets.

See notes in Table 9. In Kuroda et al. (2009), no cluster representatives are available, so the cluster member with the best resolution was arbitrarily selected in each case, in order to identify the correspondences with the results from the present study.

This work
[CDR-L3 cluster]	Martin & Thornton, 1996
〈corresponding cluster/canonical〉
(level-3 of external median)
(corresponding members)	North, Lehmann & Dunbrack, 2011
〈corresponding cluster〉
(level-3 of external median)
(corresponding members)	Kuroda et al., 2009
〈corresponding cluster〉
(representative)
(level-3 of external representative)
(corresponding members)	
L3-5-I	–	–	–	
L3-7-I	7A/4 (L3-7-I-1-2) (1/1)	L3-7-1
(L3-7-I-1-2) (2/2)	4(1MIM)
(L3-7-I-1-1) (1/1)	
L3-8-I	8B/- (L3-8-I-1-1) (1/1)	L3-8-1
(L3-8-I-1-1) (14/15)	3B(1PZ5)
(L3-8-I-2-1) (4/4)
6(1Q9W)
(L3-8-I-1-1) (6/6)	
L3-8-II		L3-8-cis6-1
(L3-8-II-2-1) (3/3)	7(2FAT)
(L3-8-II-2-1) (2/2)	
L3-8-III	8A/3 (L3-8-III-1-1) (1/1)	L3-8-2
(L3-8-III-2-1) (3/4)	3A(1YQV)
(L3-8-III-1-1) (2/2)	
L3-8-IV	–	–	–	
L3-8-V	–	–	–	
L3-8-VI	–	–	–	
L3-9-I	9A/1 (L3-9-I-1-1) (40/40)	L3-9-cis7-1
(L3-9-I-1-1) (219/219)
L3-9-2
(L3-9-I-9-1) (12/12)
L3-9-cis7-2
(L3-9-I-15-2) (8/8)
L3-9-cis7-3
(L3-9-I-12-4) (2/2)	1(1MJU)
(L3-9-I-1-2) (159/161)	
L3-9-II	9C/4λ (L3-9-II-1-8) (2/2)
9D/- (L3-9-II-1-4) (2/2)
9E/1 (L3-9-II-5-1) (1/1)	L3-9-1
(L3-9-II-2-1) (17/22)	1A (1A6V)
(L3-9-II-1-4) (5/5)
1B (7FAB)
(L3-9-II-1-8) (1/1)
1C (1Q0X)
(L3-9-II-2-2) (2/2)	
L3-9-III	9B/2 (L3-9-III-1-1) (1/1)
9F/- (L3-9-III-7-1) (1/1)	L3-9-cis6-1
(L3-9-III-1-1) (1/1)	(9-)2 (2FBJ)
(L3-9-III-1-1) (1/1)	
L3-9-IV	–	–	–	
L3-9-V	–	–	–	
L3-9-VI	–	–	–	
L3-10-I	–	–	–	
L3-10-II	–	–	–	
L3-10-III	–	L3-10-1 (L3-10-III-1-2) (2/6)	–	
L3-10-IV	–	L3-10-cis7,8-1 (L3-10-IV-1-2) (1/1)	5(1JGU) (L3-10-IV-1-2) (1/1)	
L3-10-V	–	–	–	
L3-10-VI	–	–	–	
L3-10-VII	10B/-(L3-10-VII-3-1) (1/1)	–	–	
L3-10-VIII	–	–	–	
L3-10-IX	–	–	–	
L3-10-X	–	–	–	
L3-10-XI	–	L3-10-cis8-1 (L3-10-XI-1-2) (1/2)	–	
L3-10-XII	10C/- (L3-10-XII-3-1) (1/1)
10D/- (L3-10-XII-8-1) (1/1)	–	–	
L3-11-I	11A/5λ (L3-11-I-1-1) (2/2)	L3-11-1 (L3-11-I-1-2) (8/9)	(11-)2 (2FB4) (L3-11-I-1-1) (3/5)	
L3-11-II	–	L3-11-cis7-1 (L3-11-II-1-2) (1/1)	8(2NY1) (L3-11-II-1-2) (1/1)	
L3-11-III	–	–	–	
L3-11-IV	–	–	–	
L3-11-V	11B/- (L3-11-V-1-1) (1/1)	–	–	
L3-11-VI	–	–	–	
L3-11-VII	–	–	–	
L3-11-VIII	–	–	–	
L3-11-IX	–	–	–	
L3-12-I	–	–	–	
L3-12-II	–	L3-12-1 (L3-12-II-1-1) (1/1)	–	
L3-12-III	–	–	–	
L3-12-IV	–	–	–	
L3-13-I	–	L3-13-1 (L3-13-I-1-1) (1/3)	–	
L3-13-II	–	–	–	
L3-13-III	–	–	–	
Outliers				
L3-10-O	10A/5 (L3-10-O-6-1) (1/1)	–	–	

Table 12 Comparison of level-1 conformational clusters obtained in CDR-H1 with external sets.

See notes in Table 9. In Martin & Thornton (1996), the CDR extents definition is significantly different (H26-H35), but correspondences based on median structures are shown for reference (in square brackets and full-italics).

This work
[CDR-H1 cluster]	Martin & Thornton, 1996
〈corresponding cluster/canonical〉
(level-3 of external median)
(corresponding members)	North, Lehmann & Dunbrack, 2011
〈corresponding cluster〉
(level-3 of external median)
(corresponding members)	
H1-10-I	–	H1-10-1 (H1-10-I-1-2) (2/2)	
H1-13-I	[10A/1 (H1-13-I-1-2) (43/44)]	H1-13-1 (H1-13-I-1-1) (261/267)
H1-13-2 (H1-13-I-13-4) (2/7)
H1-13-4 (H1-13-I-2-19) (3/4)
H1-13-7 (H1-13-I-8-4) (3/3)	
H1-13-II	–	H1-13-8 (H1-13-II-4-1) (2/3)	
H1-13-III	–	H1-13-6 (H1-13-III-1-2) (2/4)
H1-13-cis9-1 (H1-13-III-2-4) (2/2)	
H1-13-IV	–	–	
H1-13-V	–	–	
H1-13-VI	–	–	
H1-13-VII	–	–	
H1-13-VIII	–	H1-13-5 (H1-13-VIII-1-5) (4/4)	
H1-13-IX	–	–	
H1-13-X	–	–	
H1-13-XI	–	–	
H1-14-I	[11A/2 (H1-14-I-11-1) (1/1)]	H1-14-1 (H1-14-I-3-11) (11/11)	
H1-15-I	[12A/3 (H1-15-I-2-7) (1/1)]	H1-15-1 (H1-15-I-2-3) (9/9)	
H1-15-II	–	–	
H1-15-III	–	–	
H1-16-I	–	–	
Outliers			
H1-12-O	–	H1-12-1 (H1-12-O-1-1) (1/1)	
H1-13-O	[10B/1 (H1-13-O-66-1) (1/1)
10C/1 (H1-13-O-20-3) (1/1)
10D (H1-13-O-31-1) (1/1)]	H1-13-3 (H1-13-O-14-1) (5/5)
H1-13-9 (H1-13-O-57-1) (1/3)
H1-13-10 (H1-13-O-34-1) (2/2)
H1-13-11 (H1-13-O-56-1) (1/2)	
H1-16-O	–	H1-16-1 (H1-16-O-1-1) (1/1)	
H1-24-O	–	–	

Table 13 Comparison of level-1 conformational clusters obtained in CDR-H2, with external sets.

See notes in Table 9.

This work
[CDR-H2 cluster]	Martin & Thornton, 1996
〈corresponding cluster/canonical〉
(level-3 of external median)
(corresponding members)	North, Lehmann & Dunbrack, 2011
〈corresponding cluster〉
(level-3 of external median)
(corresponding members)	
H2-8-I	–	H2-8-1 (H2-8-I-1-1) (2/2)	
H2-9-I	9A/1 (H2-9-I-1-1) (8/8)	H2-9-1 (H2-9-I-1-1) (76/77)
H2-9-3 (H2-9-I-3-2) (2/2)	
H2-9-II	–	H2-9-2 (H2-9-II-1-2) (2/2)	
H2-9-III	–	–	
H2-9-IV	–	–	
H2-9-V	–	–	
H2-9-VI	–	–	
H2-10-I	10A/2 (H2-10-I-1-6) (17/21)	H2-10-1 (H2-10-I-1-3) (151/155)
H2-10-6 (H2-10-I-5-1) (2/3)	
H2-10-II	10B/3 (H2-10-II-1-4) (11/11)	H2-10-2 (H2-10-II-1-1)(40/42)
H2-10-4 (H2-10-II-4-1) (7/7)
H2-10-5 (H2-10-II-3-1) (3/3)	
H2-10-III	–	–	
H2-10-IV	–	–	
H2-10-V	–	–	
H2-10-VI	–	–	
H2-10-VII	–	–	
H2-10-VIII	–	–	
H2-10-IX	–	–	
H2-10-X	–	–	
H2-12-I	12A/4 (H2-12-I-5-1) (2/2)
12B/4 (H2-12-I-1-11) (2/2)	H2-12-1 (H2-12-I-1-1) (26/26)	
H2-12-II	–	–	
H2-12-III	–	–	
H2-12-IV	–	–	
H2-15-I	–	H2-15-1 (H2-15-I-1-1) (1/1)	
H2-15-II	–	–	
Outliers			
H2-10-O	10C/3 (H2-10-O-20-1) (2/2)
10D/2 (H2-10-O-36-1) (1/1)
10E/2 (H2-10-O-34-1) (1/1)
10F/2 (H2-10-O-11-2) (1/1)	H2-10-3 (H2-10-O-3-10) (10/11)
H2-10-7 (H2-10-O-20-1) (2/2)
H2-10-8 (H2-10-O-13-1) (1/2)
H2-10-9 (H2-10-O-29-3) (2/2)	

Rogue clusters and sequences

Assigned as ‘rogue’ were two conformational clusters that contain one or more members with identical CDR sequences. This definition was first used for CDR conformations by Martin & Thornton (1996) with respect to their unpredictability by canonical sequence templates when all their key residues are overlapping. In this work there is an expansion of this notion with the term ‘rogue CDR sequences’. This refers specifically to those identical sequences that are found to exist with more than one distinct conformation. The extraction of such sequences allows for further investigation, which can reveal any particular circumstances or neighbouring sequence features that led to a different CDR conformation despite the identical sequence. For example, examination of antibody Fvs with rogue CDR sequences may reveal the influence of neighbouring main-chain atoms, a particular framework residue influencing the CDR conformation, a conformational switch due to interface interactions (e.g., with an antigen), intrusive crystal-packing interactions, or even suggest some experimental error.

All cluster populations were parsed for rogue CDR sequences and a list of CDRs, tagged by their cluster assignment, was created for future detailed analysis (Supplemental Information 1). Also in the same file, entries with completely identical Fvs which belong to different conformational clusters (full-chain rogues) are reported separately, while entries containing bound antibodies are flagged as such by an asterisk. Furthermore, cluster populations were compared in all CDR/length sets, and the minimum number of amino acid differences, position-by-position, was calculated between any two sequences of different clusters. This difference was termed the ‘minimum pairwise Sequence Distance between clusters’, or mSD (essentially a minimum Hamming distance between sequences). Matrices showing the mSD between all clusters were constructed for every CDR/length, and heatmaps were produced in order to allow a quick visual appreciation of the degree of sequence dissimilarity between clusters (Supplemental Information 3). The purpose of these heatmaps is to assist mutation studies by promptly directing the researcher to clusters/CDR sequences of interest, as well as sequence-to-structure studies by biologists or modellers.

Investigation of structure resolution in outlier space

As a preliminary layer of quality assessment for the outliers in the present clustering, the min, max, average and median resolutions were calculated in clustered and outlier spaces per CDR/length (-L1, -L3, -H1, -H2, being of the highest interest). These values were plotted as stock charts for comparison, in order to observe any global correlation between the outlier space content and possibly erroneous CDR structures due to poor resolution (Supplemental Information 7). In only four cases (CDR-H1/15-, CDR-H1/16-, CDR-L1/12- and CDR-L1/16-residues) was the median resolution of outlier space found to be more than 0.5 Å higher than the respective median in clustered space, and in only two cases (CDR-H1/15-residues, 3 outliers in total, and CDR-L1/12-residues, 1 outlier in total) was the outlier median resolution value above 2.8 Å. In conclusion, average structure resolution does not appear to be a determinant factor of the outlier content, although it remains possible that wrong structures due to poor resolution may exist between the outliers. In fact, as proposed throughout this work, any decisions on structure validity should be considerably easier to make during targeted analysis of the structures/clusters of interest, when using the results of the present clustering. The supplementary file (Supplemental Information 7) also contains complementary bar charts showing the percentages of bound content in outlier and clustered space.

Discussion

The early approach to CDR conformational classification defined a strict threshold of similarity for clusters, beyond which any new conformation becomes the first member of a new class/cluster. As the number of new antibody structures increased almost exponentially in the past decades, the definition of a strict similarity threshold became problematic as many conformational variants of known classes appeared in the similarity-criterion space between different clusters. An obvious solution to this new and complex data structure was the pre-exclusion of all structures with characteristics that could potentially point to wrong conformations, or essentially be characterised as “noise” in the data. For instance, in the latest CDR clustering (North, Lehmann & Dunbrack, 2011), the data was considerably simplified by removing structures based on several filtering criteria: crystal resolution; high CDR backbone, or non-reported B-factors; presence of cis-peptide bonds for residues other than a proline; highly improbable backbone conformations and loops with very high conformational energies. In the present study however, the goal was set to obtain a classification for every available CDR, so any “data noise” had to be handled by the clustering methodology.

The primary characteristic of the CDR clustering performed in this study is that the main, or level-1, clusters do not carry a pre-defined degree of conformational similarity. This would require the strict definition of a threshold in the RMSD distance on all Cα-atoms from the cluster’s medoid, or as a maximum cluster diameter (e.g., Martin & Thornton, 1996; Kuroda et al., 2009). Alternatively in North, Lehmann & Dunbrack (2011), a dihedral angle-based distance measure was used in order to define a threshold for cluster merging (65° between each dihedral pair), while the main clustering method (an affinity algorithm) practically produced a final result that is roughly equivalent or close to the level-2 clustering in this study (clustering by r-RL). In contrast in this study, level-1 clusters were formed with no use of discreet distance thresholds whatsoever, but instead based on the greater affinity of each object towards its assigned cluster as expressed by the all-positive SWs; while the average SC ensured a typically textbook-defined, reasonable or better global partition of clusters (SC ≥ 0.51).

This approach was selected for two reasons: (1) in order to reduce the subjectivity that is inherent with every threshold definition and clustering decision in general, and (2) in order to allow the adherence of conformational variants to their most apparent closest conformational theme. This in turn may reveal the natural flexibility in physiological conditions, or structural mechanisms and synergies that are specific to an antibody’s function. Indeed, it becomes more straightforward to comparatively examine the reason for a conformational variant when it is found connected to its closest conformational theme, rather than when treated as a completely distinct conformation or as an outlier/singleton. This is also the most important difference between the present antibody CDR clustering analysis and the clustering by UPGMA offered by the recently released CDR structural database SAbDab (Dunbar et al., 2014).

The clustering algorithm employed in this study offered simultaneous flexibility in selecting the most appropriate pruning parameters, and in-depth description of clusters by its definition of cluster core objects. Researchers wishing to retrieve the most representative objects (the most tightly represented conformation) of each cluster may select any one of the cluster’s core CDRs (tagged as such in the clustering results listings). Furthermore, the presentation of each cluster’s extremities in the results (most distant members forming the cluster’s diameter), allows the rapid assessment of the extents of conformational variability of the cluster so that researchers can make informed decisions as to the importance of any observed deviations of their target structure with regard to the overall conformational characteristics of the cluster.

In practice over 80% of the clustering was straightforward in establishing a partition with an SC ≥ 0.51, all positive individual SW, the highest number of clusters possible with close-to-ideal maximum diameters and the lowest number of outliers. In fact, the formalisation of the complete procedure contains few subjective features, namely those of the ideal maximum cluster diameter index and of the overall stringency in examining all possible outcomes (average and complete hierarchical trees, 2nd-stage PAM). In the first case, the index had a merely suggestive role in triggering the assessment of a possible cluster splitting strategy, while in the second case the optional PAM stage or one of the two hierarchical methods may be completely omitted, especially if an acceptable result is already obtained. Therefore, this clustering method can be entirely machine-coded and carried out in a fully automated way, if required.

The major challenge in this clustering was brought by the initial decision to include all the available antibody structures as of the 31st December 2011 edition of PDB, in order to create a complete CDR conformational repertoire. While this decision allowed a richer result, and for all the reasons and possible advantages detailed earlier, it was accepted that noise was added to the dataset by the inclusion of a number of potentially erroneous structures. The usual strategy followed in such cases is data re-sampling, or bootstrapping, in order to assess the effects and influence of noise to the dataset configuration by some estimator (e.g., percentiles, medians, variance, etc.) and to attempt projections for the evolution of partitions in the future. There was reluctance in pursuing such a methodology in this case, mainly because the appearance of new antibody structures in the PDB follows a constantly varying scientific interest for diseases, therapeutics and basic research, and as such the obtained dataset cannot be considered representative of some random process. In this sense it is anecdotal that a few months before the closure of the dataset, a considerable number of anti-HIV and anti-‘flu antibody structures (33/128 structures released in 2011, i.e., ∼26%), all with very characteristic CDR conformations, had emerged in the PDB following the research trend for that period.

The solution to noise data was the efficient exclusion of outliers/singletons from clusters, coupled with the nested architecture of the final clustering result. The efficient exclusion was ensured by the requirement that clusters form a tight core while all cluster objects present an individual positive SW with respect to the global cluster partition. Though it was still possible that few, very small 2- or 3-member clusters failed to form due to the positive SW requirement, the subsequent 2nd- and 3rd-level qualitative clustering, based on Ramachandran Logos, would create a common conformational tag to allow recognition and classification of even such small outlying groups. Daughter-level sub-clusters mainly provide a means to identify all the members of important or subtle conformational variants of the parental theme, and by that fact offer more common examples for the researcher to compare their CDR with. Finally, it remains the individual researcher’s decision as to which CDR conformations are useful, important, or potentially wrong. However when consulting the clustering results of this study, the data is classified in such a way and with no loss of information due to pre-filtering, that the researcher has at their disposal all the necessary information to help them take that decision.

As a means of external validation, it is important to observe the comparison and relation of conformational CDR clusters between this and the major previous studies. As far as the first five CDRs are concerned, in many cases clusters from previous work were found to correspond to level-1 clusters from this study on a one-to-one basis (36/72 compared clusters from North, Lehmann & Dunbrack (2011), 21/49 compared clusters from Martin & Thornton (1996), 8/13 compared clusters from Kuroda et al. (2009)), while in several cases more than one cluster from those external sets was found to correspond to the same level-1 cluster (correspondingly for the aforementioned studies: 25/72 clusters contained in 9 level-1 clusters, 15/49 clusters contained in 7 level-1 clusters, and 5/13 clusters contained in 2 level-1 clusters). This is characteristic of the different clustering strategies adopted in each study, as the external sets imposed discreet similarity thresholds on their cluster definition, but also of the fewer number of structures in their datasets which allowed for a sharper, more specific clustering when the data configuration was favourable. In all those cases, the external clusters are still distinct in the present clustering result, as they almost always correspond to different level-2 clusters from this study. In only two cases (clusters 16A/16C in CDR-L1 from Martin & Thornton (1996), and clusters 1A/1B in CDR-L3 from Kuroda et al. (2009)) were external clusters differentiated only at the 3rd-level, meaning that the full, 3-level conformational logo is required to describe them. Finally, in several cases small 2-, or 3-member external clusters, or mere singletons, were found to correspond to outliers in this study (11/72 in North, Lehmann & Dunbrack (2011), 13/49 in Martin & Thornton (1996)), because of the specific requirements for the existence of a tight core and all positive individual SW, as explained previously. Even so, these small external clusters are still distinct in the present result as their members are regrouped at the 2nd-level of clustering. The additional full population analysis of cluster assignments between this study and previous work showed consistency of membership correspondences, at 98% (262/268) for Martin & Thornton (1996), at 97% (1,534/1,589) for North, Lehmann & Dunbrack (2011) and at 98% (188/192) for Kuroda et al. (2009). Most of the observed discrepancies concerned outlying conformations (6/6, 32/55 and 2/4, correspondingly for the aforementioned works). In comparison, the present clustering analysis revealed 117 level-1 clusters in the first five CDRs, 66 of which have no correspondences and are novel. This is due to the larger dataset and to the lack of data pre-filtering.

In CDR-H3, full population correspondences with North, Lehmann & Dunbrack (2011) were expectedly poor (56%, 171/307). This is explained by the much larger number of clustered structures (2,032 versus 307) and the different strategy employed in level-1 cluster formation, but also to some extent, by the discrepancy of 2 residues in the respective CDR-H3 definitions. Indeed, the inclusion of all available CDR-H3 loops in the present clustering procedure allowed an even clearer appreciation of their pronounced conformational hypervariability: 25 H3 lengths, 213 clusters, most of which are in fact singletons that technically acquired the status of a ‘cluster’, because they were represented by more than one structure in the initial dataset. In fact, only 53/213 clusters were populated by more than 1 unique CDR sequence; while a revealing total of another 412/2,032 structures were left as outliers/singletons. In this landscape of variability in conformation, sequence and length, the adopted level-1 clustering methodology doesn’t expand a cluster’s radius towards closely-related conformations, but instead restricts that radius appropriately, excluding structures that both fail to form a well-separated core and do not clearly belong to one cluster rather than another. However, these outlying structures are still further classified based on their Ramachandran logos, whenever possible (i.e., at level-2 and -3 of the classification scheme).

All these observations are suggestive of the advantages brought by the multi-level clustering structure, as nearly all identified external clusters are distinct at the 2nd-level of our clustering (mainly in the first five CDRs), with the 1st-level expanding towards closely-related conformational variants when possible, while efficiently excluding outliers. 3rd-level clusters procure even deeper specificity when required. It becomes apparent that the trade-off between conformational specificity and sensitivity is locked in the clusters of previous studies based on the existence of a strict, but subjective, formation threshold. In contrast, the present clustering result produced a more adaptable framework, where the sensitivity and specificity of conformational similarity are more intuitively distributed in its three different levels. As an example of the conformational variability between level-1 clusters in this study and North, Lehmann & Dunbrack (2011), a comparative view of all detected clusters in CDR-H1 13-residues (displaying a rich cluster repertoire) superposed on those from North, Lehmann & Dunbrack (2011) where applicable, is presented in Fig. 7.

Figure 7 A comparative view of all CDR-H1/13 residue clusters obtained in this work (in yellow), superposed to their correspondences from North, Lehmann & Dunbrack (2011), where applicable. Level-1 clusters from this work expand whenever possible towards closely-related variants, which are then further classified at levels 2 and 3 (complete 3-level classification in this work of the external median is given in brackets). This can be appreciated in clusters H1-13-I and H1-13-III from this work. The last four structures of this figure correspond to cluster medians from North, Lehmann & Dunbrack (2011) that were classified as outliers/singletons in this work.

The description and commenting of each CDR/length combination obtained in this study may be of small value at this point, firstly due to the massive volume of the data involved, but mainly because the detailed examination of each cluster could warrant a separate, dedicated study in its own right (something that the present study aims to assist and encourage). Nonetheless, it is interesting to observe that in almost all CDR/length combinations with substantial content in unique CDR sequences (i.e., more than 10 unique sequences) there is usually a single cluster which regroups the large majority of the available known conformations, while the remaining fraction may be populating a considerable number of much smaller clusters. In the 15 lengths (first 5 CDRs) that contained more than 10 unique sequences in their clustered population and produced more than one cluster, the major cluster of each length represented on average 74% of the available unique sequences (median: 86%). The case of H2/10-residues is the one exception with two well-populated clusters (H2-10-I and -II) with an approximate 1:2.5 ratio in non-redundant members. L3/10-residues are the only other exception where no major cluster is observed despite the considerable amount of available unique sequences.

Given the considerable volume of structural data included in the work, the above fact could be suggesting that in contrast to the original observation that CDRs adopt one of a limited number of possible conformations in L1, L2, L3, H1 and H2, in fact three out of four CDR sequences seem to result in variants of the prominent conformation for that CDR length. To take this matter even further and based on the respective median, it can also be inferred that in half the well-populated CDR lengths, a variant of the prominent conformational theme is adopted by close to nine out of ten CDR sequences. Furthermore, the animal sources of CDR members of these major clusters are sufficiently varied to suggest that the respective conformations are ubiquitously maintained. These observations combined highlight the importance of subtle conformational variations in antigen recognition and, therefore, of the detailed repertoire provided at levels 2 and 3 of the present clustering analysis (e.g., by rogue analysis at the daughter cluster level). In contrast, the hypervariable (in length, sequence and conformation) CDR-H3 appears as the loop that consistently confers the most pronounced layer of conformational variation in the antibody binding interface.

It is known from experience with humanised antibodies (Saldanha, 2009) that the conservation of residues which maintain the conformation of the CDR in the designed sequence often leads to binding versions and vice versa. Further investigation of the clusters, particularly at levels 2 and 3, for these residues will enhance the modelling and design of humanised sequences by recognition, within the variants, of subtle differences to the main conformational theme.

Conclusion

By producing a classified snapshot of the entirety of the CDR conformations in the PDB, the aim was to present the experimentally known repertoire in a way that also allows inferences on the relationship between conformations. The latter exist as the result of backbone flexibilities, induced-fit, local sequence causing subtle variants, or even erroneous experimental data. Consequently, any conclusions on the quality or truthfulness of a structure can be drawn by the aid of this classification, instead of arbitrarily discarding all dubious cases from the very beginning. The dedicated analysis of structures belonging to different clusters, despite having the same CDR or even complete Fv sequence, could prove helpful towards this end. Therefore, the present clustering study can be viewed as a necessary ‘logistical task’, where no information is lost, whose value is best described by the possibilities it offers for a range of future specialised analyses, rather than a ‘one-stop’ study that allows derivation of final conclusions on the available CDR conformations. The results provided here include richly annotated cluster summaries and cluster memberships, a three-level classification, detailed comparisons with previously established CDR conformational clusters, lists of rogue CDR sequences and minimum Sequence Distance heatmaps.

The focus of this study was to produce a complete repertoire of available CDRs, with multi-level clusters that allow the user to select the desired conformational specificity or sensitivity, but also with an increased potential for predictability from sequence. As a piece of subsequent work based on the present clustering results, a comparative assessment of predictive methods from sequence of CDR conformation (canonical templates, sequence rules and a new method named Disjoint Combinations Profiling (DCP)) was carried out by the same group (Nikoloudis, Pitts & Saldanha, 2014), with very encouraging results. An implication that could be attributed to those results, considering that no clustered data was discarded, is that the present clustered set was conformationally meaningful at its level-1 instance, despite the designed tendency of clusters to expand towards potential variants of the main conformational theme. This is based on the fact that using this clustered set for training/updating produced DCP models achieving a range of 90%–99% cumulative accuracy on predictable conformations of the new dataset (CDR-L1, -L3, -H1, -H2, -H3-base), while canonical templates achieved 91% and 94% in CDR-L1 and CDR-L3, respectively. Therefore, the clustering goal of presenting a complete repertoire of conformational families could be considered successful as the most related backbone variations were attributed correctly to the most appropriate class. This clearly did not negatively influence class identification from sequence and possibly even enhanced it. Additionally, this companion article also includes a visual analysis of CDR structures that fall into different conformational classes despite being present in identical Fv sequences.

In conclusion, an accurate CDR classification is presented with novel characteristics, richly annotated and post-analysed clustered data, and also compared with previous work. In all cases, it is believed that the present analysis fills a gap in antibody CDR studies, by creating links between all related prior knowledge, while proposing new directions for future research.

Supplemental Information

Supplemental Information 1 Collection of rogue CDR sequences

Rogue CDR sequences in every CDR/length, with the respective lists of level-1 cluster-tagged CDRs, in which they are identified. Entries with an asterisk indicate bound structures. Additionally, entries with completely identical Fvs that belong to different conformational clusters (〈***full-chain rogues〉), are given immediately after the detected rogue CDR sequences, when applicable.

Click here for additional data file.

Supplemental Information 2 Comparison of level-1 conformational clusters obtained in CDR-H3 with North, Lehmann & Dunbrack (2011)

The cluster medoid/median or representative of the external sets was used for identification of correspondences. Only level-1 clusters with a correspondence are shown here, in order to preserve a readable size for the table (213 total level-1 clusters in CDR-H3). In brackets, next to each correspondence, is the full, level-3, classification in this work of the representative of the external set. The entire correspondence is marked between square brackets and in full-italics because the CDR-H3 definition used in North, Lehmann & Dunbrack (2011), was longer by 2 residues (i.e., 93-102).

Click here for additional data file.

Supplemental Information 3 Minimum sequence distance (mSD) heatmaps

Collection of heatmaps for all CDR/length combinations, showing the minimum number of amino acid differences, position-by-position, between any two sequences of different clusters. allow a quick visual appreciation of the degree of sequence dissimilarity between clusters. mSD heatmaps allow a quick visual appreciation of the degree of sequence dissimilarity between clusters.

Click here for additional data file.

Supplemental Information 4 Summary for the clustering of CDR-H3

Level-2 clusters are shown exceptionally (marked with an asterisk) when no level-1 cluster is formed (minimum of 2 members required).

Click here for additional data file.

Supplemental Information 5 Detailed membership assignments (sorted by cluster)

Csv formatted lists where every CDR is shown in cluster order with all available clustering and data-mined information (cis/trans peptides, structure resolution, crystal spacegroup, sequence, Ramachandran logos, cluster core label, bound state, light isotype, heavy-only and light-only labels).

Click here for additional data file.

Supplemental Information 6 Detailed membership assignments (sorted by PDB order)

Csv formatted lists where every CDR is shown in alphabetical PDB order with all available clustering and data-mined information (cis/trans peptides, structure resolution, crystal spacegroup, sequence, Ramachandran logos, cluster core label, bound state, light isotype, heavy-only and light-only labels).

Click here for additional data file.

Supplemental Information 7 Resolution and bound-state comparative charts between clustered and outlier space

Average resolution values plotted as stock charts for comparison, in order to observe any global correlation between the outlier space content and possibly erroneous CDR structures due to poor resolution. Additionally, bar charts show the percentages of bound content in outlier and clustered space.

Click here for additional data file.

Additional Information and Declarations

Competing Interests

Author Contributions

The authors declare there are no competing interests.

Dimitris Nikoloudis conceived and designed the experiments, performed the experiments, analyzed the data, contributed reagents/materials/analysis tools, wrote the paper, prepared figures and/or tables, reviewed drafts of the paper.

Jim E. Pitts contributed reagents/materials/analysis tools, reviewed drafts of the paper, expert advice, general project supervision.

José W. Saldanha conceived and designed the experiments, analyzed the data, contributed reagents/materials/analysis tools, wrote the paper, reviewed drafts of the paper, expert advice, general project supervision.

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
