# Peer review of "A complete, multi-level conformational clustering of antibody complementarity-determining regions"

_PeerJ, doi:10.7717/peerj.456_

## Round 0.1 · original submission · Minor Revisions

As you have submitted two linked manuscripts, I have endeavoured to have the same reviewers assigned to both manuscripts. The reviewers agree that your work warrants separate manuscripts, yet provided some comments for you to address for each separate manuscript.

For this manuscript, the comments are minor, and you can find the details below. You will shortly receive my decision on the linked manuscript as well.

·

Basic reporting

The article is clearly written and free of typographical errors.

Experimental design

The experimental design is clearly explained and well implemented. The novel idea to use all the available data is well justifed and commented upon. The data mining involving clustering and pruning of dendrograms is state of the art.

Validity of the findings

The data are robust and statistically sound. The paper presents a new classification of recognition loops on antibodies that uses all the available data (up to 2011) and should be of real use to researchers engaged in understanding or designing antibodies.

·

Basic reporting

The manuscript ‘A complete, multi-level conformational clustering of antibody complementarity-determining regions’ of Nikoloudis and coworkers describes a machine learning based method to group/determine the different canonical states of the CDRs of Abs. The manuscript is fluently written.

Experimental design

No comments.

Validity of the findings

The authors decided to not remove redundant Abs. This is to my opinion a very good approach, because indeed, this allows the usage of much more structural information (Ab can be unbound in one structure and bound to an antigen in another structure). But, a short description on whether or not these different PDB-entries fall in different classes is missing and should be added.
A short elaboration on the ‘special’ Abs should be given. Do cameloid-derived Abs full within a separate cluster? What about Bence-Jones Abs?

Additional comments

No comments.

Reviewer 3 ·

Basic reporting

In this manuscript, the authors reported classification of CDRs of antibodies based on crystal structures in the PDB. It is well known that antibodies have a limited number of conformations in the antigen-binding site other than the H3 region. So a lot of similar works to this study have been done in the last decades. However, As clearly written in the Introduction, an advantage of this work is that the authors classified redundant antibody structures, so it is most comprehensive.

Although there are a few public databases, such as SAbDab, that make it possible to analyze all available CDR structures, this is the first study that explicitly examined all the antibody structures.

Experimental design

This work is technically sound, and this manuscripts could be a dictionary for antibody engineers.

I have two comments on the flow of the analysis.

The authors classified redundant CDRs, but they did not discuss the outliers of the classifications at all. I wonder if the authors could go a step further by analyzing those outliers. For example, crystal structures that have bad resolution tend to be in the outliers? CDRs derived from a particular germline gene could be more diverse? Does antigen-binding affect canonical conformations?

Although the authors compared their results with most of the previous works and made the Tables for comparison, they missed a recent work, which also analyzed redundant CDR, but only for L3 regions (Teplyakov and Gilliland Proteins 2014). I would suggest the that authors also compare their result with it. Teplyakov and Gilliland showed the same L3 sequences could have the same canonical conformations even when they paired with different VH domains. Is this true for the other CDRs?

Validity of the findings

The data is sound, but could be better, as I mentioned above.

In the final paragraph in the page 25, the authors concluded that the classification was presented with annotated and post-analzyed clustered data, and this work filled a gap in related studies by creating links between all related prior knowledge. However, the authors did not discuss the outliers, which, I think, are most important in this work.

On the other hand, the authors themselves mentioned that this work would need a separate dedicated study for each cluster to make it more valuable (1st paragraph in the page 23). These two paragraphs sound contradicting.

Also, the authors stated that this work is useful for antibody humanization. But the following statement in the page 23 is not clear.

"the conservation of canonical residues in the designed sequence often leads to binding versions and vice versa."

Additional comments

The authors used several text searches to extract antibody structures in the PDB. This is sufficient. But, probably, an easier and more accurate way might have been that they use BLAST against the whole PDB sequences.

---

## Round 0.2 · accepted · Accept

Congratulations on the acceptance of your manuscript.